# Subgoal-Guided Policy Heuristic Search with Learned Subgoals

**Jake Tuero** [1 2]  **Michael Buro** [1]  **Levi H. S. Lelis** [1 2]

## Abstract

Policy tree search is a family of tree search algorithms that use a policy to guide the search. These algorithms provide guarantees on the number of expansions required to solve a given problem that are based on the quality of the policy. While these algorithms have shown promising results, the process in which they are trained requires complete solution trajectories to train the policy. Search trajectories are obtained during a trial-and-error search process. When the training problem instances are *hard*, learning can be prohibitively costly, especially when starting from a randomly initialized policy. As a result, search samples are wasted in failed attempts to solve these hard instances. This paper introduces a novel method for learning subgoal-based policies for policy tree search algorithms. The subgoals and policies conditioned on subgoals are learned from the trees that the search expands while attempting to solve problems, including the search trees of failed attempts. We empirically show that our policy formulation and training method improve the sample efficiency of learning a policy and heuristic function in this online setting.

## 1. Introduction and Overview

This work focuses on solving single-agent deterministic search problems, using minimal domain knowledge. In particular, we are interested in "*needle-in-haystack*" problems, where finding any solution can be challenging. Many important problems can be represented as a deterministic single-agent search problem, such as robotic navigation (Tan et al., 2021) and network routing (Liu & Ramakrishnan, 2001). A recent line of research to address this class of problems is *policy tree search* algorithms (Orseau et al.,

2018; Orseau & Lelis, 2021), which use a learned policy to guide the search, *i.e.*, a probability distribution over the set of actions available to the agent. Orseau et al. (2018) showed that a benefit of using policy tree search methods over traditional heuristic search methods is that they provide an upper bound on the number of node expansions required to find a solution that depends on the *quality* of the policy, and Orseau & Lelis (2021) showed that a policy can be learned while minimizing such an upper bound.

Policy-Guided Heuristic Search (PHS*) (Orseau & Lelis, 2021) is a policy-guided search algorithm that combines a learned policy and a heuristic function. While there have been previous approaches to combining a policy and heuristic/value function, such as the PUCT-based search algorithms (Rosin, 2011; Kocsis & Szepesvári, 2006) like AlphaZero (Silver et al., 2017) and MuZero (Schrittwieser et al., 2020) in adversarial search domains like Go and Chess, Orseau & Lelis (2021) show that PHS* is better suited for the single-agent search problems we consider in this work.

Policy tree search methods, including PHS*, are usually trained online using the Bootstrap search-and-learn process (Arfaee et al., 2011). The Bootstrap process initially uses randomly initialized neural models encoding the heuristic and the policy to iteratively solve a subset of the problems. If the search cannot solve problems within a search budget, the resulting trees are discarded; if at least one problem is solved, the models are optimized on the solution trajectories found. A problem with this approach is that much of the search effort is wasted on failed attempts, especially in the early iterations of learning, with untrained neural models.

A key innovation of our approach is that we use these failed search attempts to learn subgoals, which can shorten the planning horizon and ease the learning process. These failed attempts allow us to learn policies that navigate between subgoals. Using subgoals to break down the search horizon into smaller pieces has shown success in complex domains, as demonstrated in kSubS (Czechowski et al., 2021), AdaSubS (Zawalski et al., 2022), and HIPS (Kujanpää et al., 2023). An issue with some of these approaches is that they lack completeness, *i.e.*, they might fail to return a solution even if one exists. HIPS-$\varepsilon$ (Kujanpää et al., 2024) is a variant of HIPS that is complete. However, HIPS-$\varepsilon$, as well as kSubS, AdaSubS, and HIPS, require precomputed datasets

[1]Department of Computing Science, University of Alberta, Edmonton, Canada [2]Alberta Machine Intelligence Institute (Amii), Edmonton, Canada. Correspondence to: Jake Tuero <tuero@ualberta.ca>.

*Proceedings of the 42nd International Conference on Machine Learning*, Vancouver, Canada. PMLR 267, 2025. Copyright 2025 by the author(s).

of solution trajectories, which may be prohibitively costly in complex environments.

We propose a hierarchical policy for policy-guided tree search algorithms that is complete and does not require precomputed solution trajectories. This is achieved by learning subgoal-guided low-level policies from the Bootstrap data. In our approach, a subgoal generator produces a set of subgoals for a given state encountered in the search. The low-level policy is then conditioned on each of these subgoals and on the current state, thus providing one probability distribution over actions for each subgoal. A high-level policy produces a distribution over the generated subgoals, which gives an importance weighting for each of the low-level policies. The low-level policies are then mixed, based on the weighting given by the high-level policy, to produce a final policy that guides the search. In addition, we propose a method to learn these subgoals and policies using the data provided by the tree search, even in contexts where a budget for the search causes early termination without a solution.

In experiments, we demonstrate the sample efficiency our method enables in that it requires substantially fewer node expansions to learn effective policies than other search algorithms trained with the Bootstrap algorithm in a variety of problem domains. We also show that policy tree search algorithms using our subgoal-based policy can learn how to solve problems from domains that HIPS-$\varepsilon$ cannot solve.

Our contributions can be summarized as follows. We provide a novel subgoal discovery method for policy-guided tree search algorithm that learns from data collected during search. Our method for learning policies automatically reduces complex problems into easier sub-problems, by automatically learning subgoals. Moreover, it can train neural policies from both successful and failed searches, unlike other algorithms that learn with the Bootstrap process.

## 2. Preliminaries

We define a single-agent deterministic search problem as the tuple $(\mathcal{S}, \mathcal{A}, T, s_0, \mathcal{S}_g)$, where $\mathcal{S}$ is the set of states, $\mathcal{A}$ is the finite set of actions available to the agent, $T : \mathcal{S} \times \mathcal{A} \to \mathcal{S}$ is a deterministic transition function, $s_0$ is the initial state, $\mathcal{S}_g$ is the set of goal states. The state reached after applying a sequence of actions $a_{0:t} = a_0, a_1, \ldots, a_t$ is denoted as $T(a_{0:t})$. The underlying state space of the search problem is modeled as a graph $G = (\mathcal{S}, A)$. This graph has one edge $(s, s')$ in $A$ for each $a$ in $\mathcal{A}$ such that $T(s, a) = s'$.

The algorithms we consider search over the state space by generating a tree. Let $\mathcal{N}$ be the set of nodes of the tree, with $n_0 \in \mathcal{N}$ being the root node corresponding to the initial state $s_0$. Any node in the tree can be defined as a sequence of actions from the initial state $s_0$, with the root node being the empty sequence of actions. For any node $n$ in $\mathcal{N}$, its set

of children is denoted as $\mathcal{C}(n)$, its parent as $\text{par}(n)$, and its set of ancestors excluding $n$ is $\text{anc}(n)$ and we also define $\text{anc}_*(n) = \text{anc}(n) \cup \{n\}$. The process in which a search algorithm generates the set of children for a node $n$ is called *expansion*, and we say that the node was *expanded*. The first node to be expanded is the root $n_0$, and all nodes $n$ can only be expanded if its parent $\text{par}(n)$ has been expanded. The search ends when there are no more nodes to expand or the search finds a node representing an $s$ in $\mathcal{S}_g$. The set $\mathcal{S}_g$ is not known a priori, but can be checked with a Boolean test.

The search algorithm incurs a loss of $\ell : \mathcal{N} \to (0, \infty]$ every time a node is expanded. The *path loss* $g(n)$ of a node is the sum of losses from the root node to $n$, given by $g(n) = \sum_{n' \in \text{anc}(n)} \ell(n')$. If $\ell(n) = 1$ for all nodes, then $g(n)$ is equal to the depth of the node, denoted $d(n)$.

A policy $\pi : \mathcal{N} \to [0, 1]$ assigns probabilities to sequences of actions which represents each node. It is defined recursively for a child $n'$ of node $n$ by $\pi(n') = \pi(n)\pi(n'|n)$, where the conditional probability $\pi(n'|n)$ in $[0, 1]$, $\sum_{n' \in \mathcal{C}(n)} \pi(n'|n) \leq 1$ (the inequality can occur due to state pruning during the search), and $\pi(n_0) = 1$. Thus, $\pi(n)$ is the product of probabilities along the path from the root to $n$: $\pi(n) = \prod_{n' \in \text{anc}_*(n) \setminus \{n_0\}} \pi(n' \mid \text{par}(n'))$.

### 2.1. Background

**Best-First Search**. Best-First Search (BFS) algorithms (Pearl, 1984), expand nodes with increasing *cost*. The root node is added to a priority queue sorted by a cost function. In every iteration, the cheapest node in the queue is removed and expanded; the nodes thus generated are added into the queue. A node will not be expanded if its underlying state has already been expanded. In the context of policy-guided search, the search terminates when a node representing a state in $\mathcal{N}_G$ is encountered (see Appendix A for details).

**Policy Tree Search**. LevinTS (Orseau et al., 2018) is a BFS algorithm using the evaluation function

$$\varphi_{\text{LevinTS}}(n) = \frac{d(n) + 1}{\pi(n)}. \tag{1}$$

Policy-Guided Heuristic Search (PHS*) (Orseau & Lelis, 2021) generalized LevinTS by using both a policy and heuristic function to guide the search, which can be defined as BFS using the evaluation function

$$\varphi_{\text{PHS}}(n) = \eta(n)\frac{g(n)}{\pi(n)}, \tag{2}$$

where $\eta(n)$ is the heuristic factor for node $n$ defined as

$$\eta(n) = \frac{1 + h(n)/g(n)}{\pi(n)^{h(n)/g(n)}}, \tag{3}$$

with $h(n)$ being a heuristic, which estimates the cost-to-go

to a goal state. For $g(\cdot) = 0$, we define $\eta(n) = 1$. LevinTS and PHS* are shown in Appendix B, Algorithms 2 and 3.

**The Bootstrap Process**. The Bootstrap algorithm (Arfaee et al., 2011) learns a neural policy and/or heuristic functions from search. The initial policy/heuristic is represented by randomly initialized neural network, and are improved in each iteration as follows. A search algorithm using the current neural model runs on a subset of the training problem instances with a bound on the number of node expansions allowed for each problem. Problems that are solved have their solution trajectory used to update the models, and the process continues. After every sweep through a set of training problems, a separate validation set is used to determine when to stop. If the search can solve all problems from the validation set, then the training is complete and the neural model is returned. If the search cannot solve any new problems after a sweep of the training set, the expansion budget is increased. See Appendix C for its pseudocode.

## 2.2. Problem Statement

Given a set of problem instances $K$, the objective is to solve them while minimizing the *total search loss*. Concretely, we use the formulation provided by Orseau & Lelis (2021): For a search algorithm $S$, a loss of $\ell(n) > 0$ is incurred by $S$ expanding node $n$, and the *search loss* $L(S, n)$ is the sum of individual losses $\ell(n')$ for all nodes $n'$ that have been expanded by algorithm $S$ up to and including $n$. The *total search loss* is then given by $\sum_{k \in K} L(S, n_k^*)$, where $L(S, n_k^*)$ is the search loss of algorithm $S$ while finding a solution node $n_k^*$ on the $k$-th problem in $K$. For example, if $\ell(n) = 1$ for all nodes $n$, then the objective is to solve the problem instances using as few node expansions as possible.

## 3. Method

We propose learning a subgoal-guided policy for policy search algorithms, which uses learned subgoals from a subgoal generator to guide the search through the use of policies conditioned on subgoals. In this work, we define subgoals as states from the underlying state space that the search attempts to achieve. Unlike previous policy search methods, our policy uses data from budget-bounded searches that fail to find a solution to train its neural policy.

Our method is given in Figure 1. Figure 1.a shows how the low-level policy, high-level policy, heuristic, and subgoal generator models interact when expanding a node during the search (Sections 3.1 and 3.2). During the Bootstrap process used to train the policy, the heuristic, and the subgoal generator models, the budgeted search will either terminate without a solution or with a solution trajectory. When the budgeted search terminates without a solution found, Figure 1.b shows how the search tree can be used to gener-

ate subgoals and trajectories between them for the subgoal generator and low-level policy to learn from, respectively (Section 3.3). Finally, Figure 1.c shows how we use solution trajectories to train the heuristic model, low-level policy, high-level policy, and subgoal generator (Section 3.4).

### 3.1. Subgoal-Guided Policy Search

Instead of a single policy $\pi(\cdot)$ that produces a probability distribution over actions, we employ several subgoal-conditioned low-level policies together with a high-level policy over subgoals. The low-level policy $\pi_\theta^{\text{low}}(a|s, \hat{s}_{g_i})$ is conditioned on a state $s$ and a subgoal $\hat{s}_{g_i}$, and represents a probability distribution over actions to achieve $\hat{s}_{g_i}$ from $s$. For $k$ subgoals, we generate a set of $k$ probability distributions by conditioning $\pi_\theta^{\text{low}}$ on each of the $k$ subgoals $\hat{s}_{g_i}$. The high-level policy $\pi_\psi^{\text{hi}}(\hat{s}_{g_i}|s)$ gives a distribution over the $k$ subgoals. These low-level and high-level policies are implemented as neural networks parameterized by $\theta$ and $\psi$, respectively.

The resulting policy $\pi^{\text{SG}}(a|s)$, which we will refer to as the *subgoal-guided policy*, is formulated by using a weighted geometric mixing of the low-level policies, in which the weight given to policy $\pi_\theta^{\text{low}}(a|s, \hat{s}_{g_i})$ is the probability for the $i$th subgoal given by the high-level subgoal policy $\pi_\psi^{\text{hi}}(\hat{s}_{g_i}|s)$:

$$\pi^{\text{SG}}(a|s) = \frac{\sum_{i=1}^{k} \pi_\psi^{\text{hi}}(\hat{s}_{g_i}|s) \cdot \pi_\theta^{\text{low}}(a|s, \hat{s}_{g_i})}{\sum_a \sum_{i=1}^{k} \pi_\psi^{\text{hi}}(\hat{s}_{g_i}|s) \cdot \pi_\theta^{\text{low}}(a|s, \hat{s}_{g_i})}. \quad (4)$$

The search algorithm then works by calling BFS with the corresponding evaluation function ($\varphi_{\text{LevtinTS}}$ or $\varphi_{\text{PHS}}$), but uses $\pi^{\text{SG}}(\cdot)$ instead of $\pi(\cdot)$.

### 3.2. VQVAE Subgoal Generator

We use a Vector Quantized Variational Autoencoder (Van Den Oord et al., 2017) (VQVAE), represented by a neural network parameterized by $\phi$, as a subgoal generator. Our design is similar to the one used in HIPS-$\varepsilon$ (Kujanpää et al., 2023). The *Encoder* takes as input a pair of states $(s_{\text{cur}}, s_{\text{tar}})$ and and produces a continuous latent code $z_e$. The latent code is quantized by finding the *nearest* entry in the *codebook* $e_c$, which is a table of $k$ vectors. The decoder takes as input the pair $(s_{\text{cur}}, e_c)$ and produces a reconstructed target $\hat{s}_{\text{tar}}$. The intuition behind the pair of current and target states given to the encoder is that the encoder learns to capture the difference between the two states into the encoding codebook vector. When the current state is given to the decoder at runtime during search, the encoding codebook vector can be used to recover the changes required to be made from the current state to reconstruct the target state.

To generate the $i$-th subgoal during search, the node's state $s$ and the $i$-th codebook entry $e_i$ of the VQVAE are given

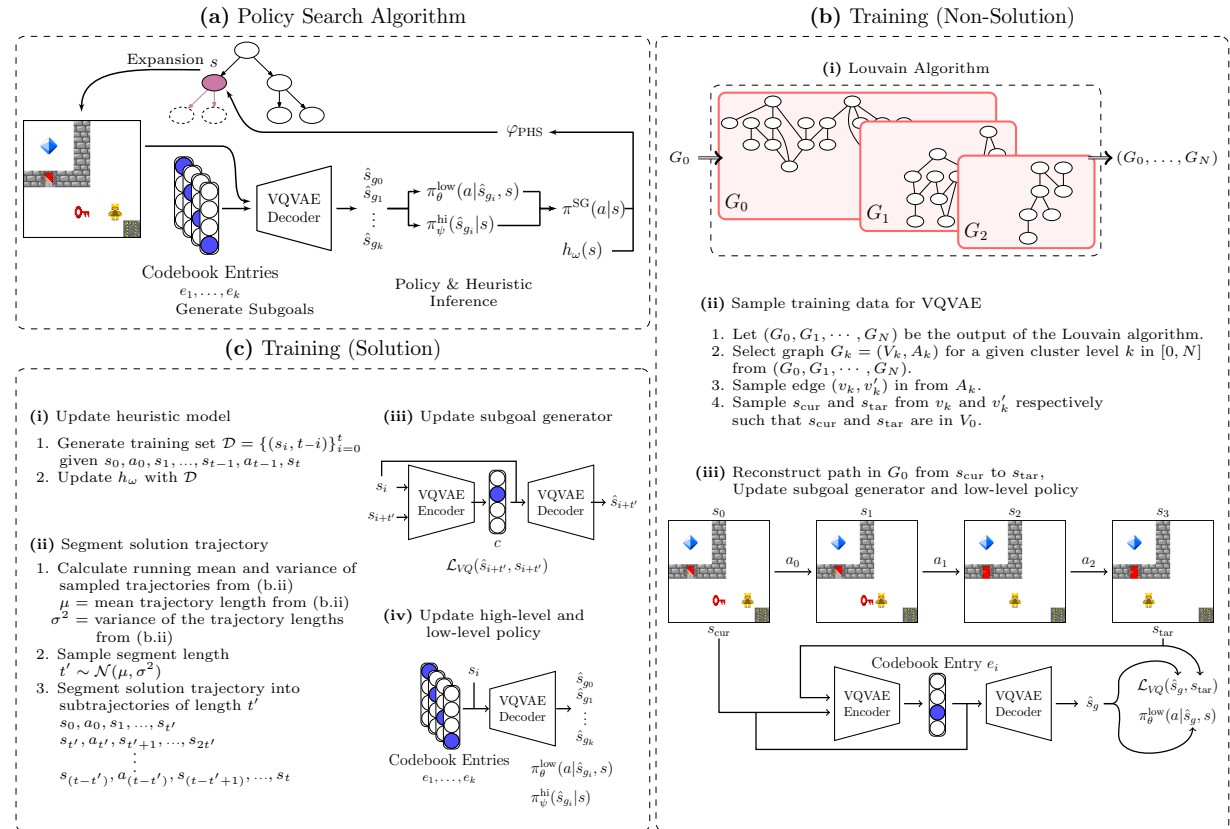

Figure 1: **(a) Tree Search**. Policy tree search generates subgoals to use with $\pi^{\text{SG}}$. **(b) Training (Non-Solution)**. **(i)** The underlying graph that induced the tree search is generated from the parent-child relationships found during search, which is used to create a hierarchy of cluster graphs using the Louvain algorithm. **(ii)** States $s_{\text{cur}}$ and $s_{\text{tar}}$ are sampled from neighbouring states in $G_i$ from the graphs created by the Louvain Algorithm. **(iv)** The resulting trajectory from $s_{\text{cur}}$ to $s_{\text{tar}}$ is used to update the subgoal generator and low-level policy. **(c) Training (Solution)**. **(i)** The heuristic is updated using the solution trajectory. **(ii)** The solution trajectory is segmented. **(iii)** The subgoal generator is updated using the partial trajectory. **(iv)** The high-level and low-level policies are updated using the segmented trajectories.

as input to the VQVAE's decoder, which produces a target subgoal state $\hat{s}_{g_i}$. The $k$ generated subgoals are then used as inputs to the low-level and high-level policies, then combined to produce $\pi^{\text{SG}}(\cdot)$, as described in Section 3.1.

The difference between our work and HIPS-$\varepsilon$ is how the VQVAE subgoal generator is used. We condition low-level policies on the generated subgoal states to guide the search to reach these subgoals, but search is performed at the search problem's action level. HIPS-$\varepsilon$ performs search at the subgoal-level, where the resulting state-transition from a node expansion comes from the generated state of the VQ-VAE. This can be problematic if the generated subgoal states are not accurate reconstructions of actual states because the reconstructed states might not be reachable by the search trying to find the subgoals. We hypothesize that HIPS-$\varepsilon$ will not be able to effectively guide the search in complex environments for which state reconstruction is *hard*. By contrast, our policy $\pi^{\text{SG}}(\cdot)$ uses the reconstructed states only

to condition the low-level policies. We further hypothesize that, in this $\pi^{\text{SG}}(\cdot)$ scheme, even imperfect reconstructions of subgoal states can be helpful in guiding the search.

To train the VQVAE subgoal generator to generate candidate subgoal states, $(s_{\text{cur}}, s_{\text{tar}})$ are encoded into the continuous latent code $z_e$, quantized to the *nearest* entry in the *codebook* $e_c$, then given to the decoder along with $s_{\text{cur}}$ to produce a reconstructed target $\hat{s}_{\text{tar}}$. A reconstruction loss $\mathcal{L}_{\text{rec}}(s_{\text{tar}}, \hat{s}_{\text{tar}})$ is used to encourage the reconstructed target subgoal state to match the given target state, along with a penalty to encourage the encoding vector to become closer to a trainable codebook entry. The full loss is denoted as $\mathcal{L}_{VQ}$, and this procedure is used to train the VQVAE in scenarios where the search terminates without solution found (Section 3.3) and when the search returns a solution trajectory (Section 3.4), with the only difference being how $s_{\text{cur}}$ and $s_{\text{tar}}$ are selected. For full details of the loss function used, see Appendix D.

### 3.3. Training From Non-Solution Search Trees

We leverage data from search trees that during the online Bootstrap training process would normally be discarded. This allows the subgoal-conditioned low-level policy and subgoal generator networks to be trained on problem instances where LevinTS and PHS* cannot find solutions. This process is shown in Figure 1.b, and in the empirical section, we show how this can lead to significant savings in the total search loss incurred during the Bootstrap training process.

Different BFS algorithms potentially expand different trees, which form different subgraphs of the underlying state space. We denote the subgraph containing the states represented by the nodes expanded in the tree as $G_0$. If the search cannot solve a problem instance during the online training process, we use this subgraph as the initial graph $G_0$ for the Louvain clustering algorithm (Blondel et al., 2008). The Louvain algorithm iteratively creates a hierarchy of graphs $(G_0, \ldots, G_N)$ through clustering (Figure 1.b.i). Each iteration of the Louvain algorithm consists of creating a hierarchical graph $G_{i+1} = (V_{i+1}, A_{i+1})$ from the current graph $G_i = (V_i, A_i)$, where $V_{i+1}$ represents a partition of $V_i$, where each $v$ in $V_{i+1}$ represents a part in this partition. The set $A_{i+1}$ contains edges $(v_{i+1}, v'_{i+1})$ for $v_{i+1}$ and $v'_{i+1}$ in $V_{i+1}$ only if there are a $v_i$ in $v_{i+1}$ ($v_i$ is in the part $v_{i+1}$) and a $v'_i$ in $v'_{i+1}$ ($v'_i$ is in the part $v'_{i+1}$) such that $(v_i, v'_i)$ in $A_i$. This process continues until a graph consisting of a single node is produced, or if $G_{i+1} = G_i$. The pseudocode of this algorithm is given in Appendix E, Algorithm 5.

We use the Louvain algorithm because Evans & Şimşek (2023) showed it to find meaningful structure in other state spaces. The difference between our method and that of Evans & Şimşek is that we learn to generalize subgoals across instances, where they assumed the state-transition graph for each individual problem was known a priori.

For a given clustering level $k$, the graph $G_k$ produced by the Louvain algorithm is selected. Two states $s_{\text{cur}}$ and $s_{\text{tar}}$ are sampled from $v_k$ and $v'_k$ respectively such that $(v_k, v'_k) \in A_k$ (Figure 1.2.ii). If the graph is directed, then a graph traversal is performed in $G_0$ between the two sampled states to determine the order of $s_{\text{cur}}$ and $s_{\text{tar}}$ such that $s_{\text{tar}}$ can be reached from $s_{\text{cur}}$ through a sequence of actions. The VQVAE subgoal generator is then trained using $(s_{\text{cur}}, s_{\text{tar}})$ as input to generate the reconstructed target subgoal state $\hat{s}_g$ of $s_{\text{tar}}$, as described in Section 3.2. The low-level $\pi_\theta^{\text{low}}$ policy is trained using the sequence of states and actions between $s_{\text{cur}}$ and $s_{\text{tar}}$ in $G_0$, while conditioned on $\hat{s}_g$ (Figure 1.b.iii).

### 3.4. Training From Solution Search Trees

If the search finds a solution, then the solution trajectory is used to train the heuristic model $h_\omega$ using the costs of the current state along the solution path to the goal state (Figure 1.c.i). Similarly to previous work, the heuristic function is learned by minimizing the mean squared error between the model's predicted value and the actual cost to go for the states along the solution path (Arfaee et al., 2011).

Instead of clustering, we segment the solution trajectory $(s_0, s_1, \ldots, s_t)$ into sub-trajectories of length at most $t' < t$ of the form $(s_i, s_{i+1}, \ldots, s_{\min(i+t',t)})$. Here, $i \in \{0, t', 2t', \cdots, \lfloor \frac{t}{t'+1} \rfloor t'\}$ and $t'$ is sampled from a Normal distribution, with mean and variance calculated from the lengths of sampled trajectories from the non-solution trees (Figure 1.c.ii). Sampling $t'$ ensures that the average number of state transitions between the $s_i$ and $s_{i+t'}$ states is close to $s_{\text{cur}}$ and $s_{\text{tar}}$ sampled in the non-solution case.

For each sub-trajectory, $s_{\text{cur}}$ and $s_{\text{tar}}$ are represented by $s_i$ and $s_{i+t'}$, respectively. Similarly to Section 3.3, the VQ-VAE subgoal generator is trained using $(s_i, s_{i+t'})$ as input to generate the reconstructed target subgoal state $\hat{s}_{i+t'}$ (Figure 1.c.iii). The low-level policy $\pi_\theta^{\text{low}}$ is trained using the sequence of states and actions $(s_i, s_{i+1}, \ldots, s_{i+t'})$ while conditioned on the subgoal $\hat{s}_{i+t'}$. The high-level policy $\pi_\psi^{\text{hi}}$ is trained by generating the $k$ subgoals following the process outlined in Section 3.2 for the state $s_i$, of which $\hat{s}_{i+t'}$ is one of them, and uses $\hat{s}_{i+t'}$ as the target for training the policy $\pi_\psi^{\text{hi}}$ (Figure 1.c.iv). All policies are trained while minimizing the Levin loss (Orseau & Lelis, 2021).

### 3.5. Analysis of $\pi^{\text{SG}}$

Policy-guided search algorithms can be implemented as BFS, but using our policy definition given in Equation 4. As a result, any policy-guided search algorithm using our policy $\pi^{\text{SG}}$ comes with all the guarantees which PHS* provides with completeness being one of them. Specifically, the requirement for completeness is that the *loss* incurred for each expansion step is non-negative, and that no infinite path has finite path loss (Orseau & Lelis, 2021). This is unlike other subgoal tree search algorithms such as kSubS, AdaSubS, and HIPS, with the exception of HIPS-$\varepsilon$.

## 4. Experiments

The goal of our experiments is to test our hypothesis that our learning process of $\pi^{\text{SG}}$ is more sample efficient than existing approaches to learning policies for search algorithms. In the extreme case, our method's sample efficiency enables policies to be learned on complex environment domains in which the existing approaches require a search budget (both in terms of the number of node expansions and in time) which are prohibitively expensive to the point where those methods cannot make any progress. We also evaluate the quality of the learned policies in terms of expansions and solution cost on a separate test set.

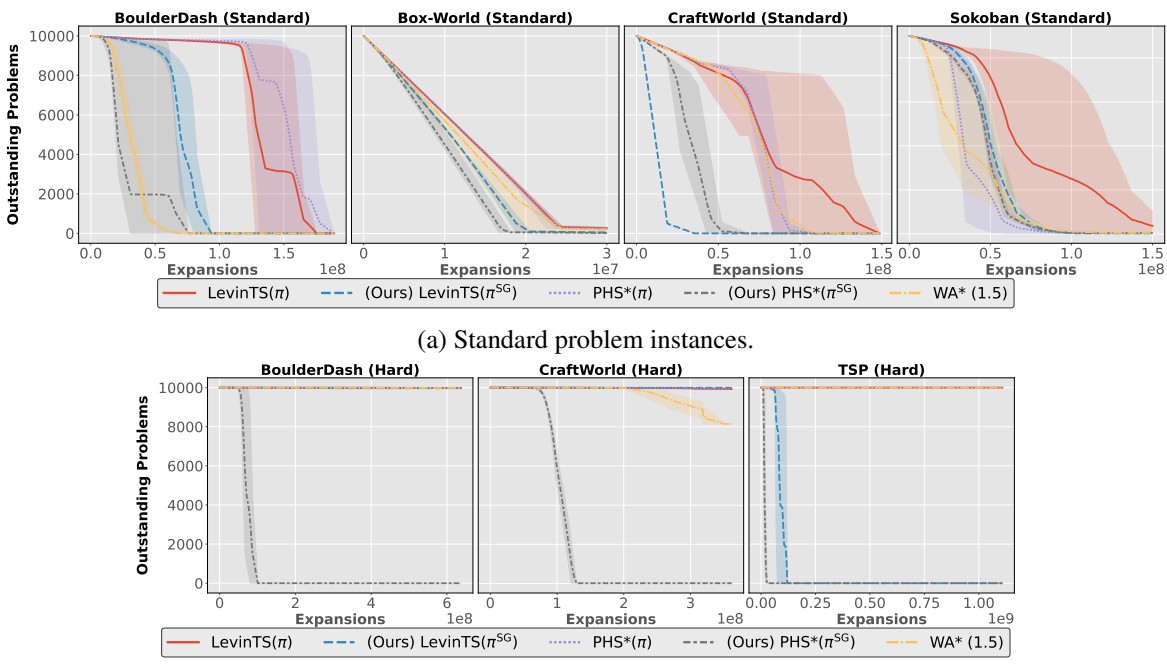

(a) Standard problem instances.

(b) Hard problem instances.

Figure 2: The line is the average number outstanding problems for the respective number of expansions accumulated during training. Shaded regions show maximum and minimum outstanding problems across all seeds.

### 4.1. Environment Domains

**Box-World**: A $20 \times 20$ room with colored keys and locked boxes (Zambaldi et al., 2018). Keys are consumed when used on a lock of the matching color, with the agent receiving a new key matching the color of the box. The objective is to open the designated goal box by opening the correct sequence of locks. If the agent opens a corresponding lock on the wrong box, the resulting scenario can become dead-end.

**CraftWorld**: A $14 \times 14$ room with various raw materials and workbenches (Andreas et al., 2017). The agent can collect raw materials to create tools. We generate problems with the open-source level generator[1] of the procedure detailed by Andreas et al. (2017).

**BoulderDash**: A $14 \times 14$ room where the agent must gather a number of diamonds to unlock the exit door, and proceed to enter it. Diamonds can be inside locked sub-rooms in which the corresponding key must be gathered first. Various dirt elements exist, which disappear once the agent walks over them, and can significantly increase the state-space size. We generate problems by randomly placing the keys, diamonds, and an exit door.

**Sokoban**: A $10 \times 10$ room where the agent must push boxes onto specified goal locations. The box can only be pushed; not pulled, so boxes can get stuck on walls of the

grid. Sokoban is PSPACE-hard (Culberson, 1999). We use the Boxoban training and test problems (Guez et al., 2018).

**Traveling Salesman Problem (TSP)**: A $10 \times 10$ grid-based environment where the agent must visit all of the cities before returning back to the first city it visited. The TSP is NP-hard (Garey & Johnson, 1979).

### 4.2. Algorithms Evaluated

**Algorithms with $\pi^{\text{SG}}$.** We evaluate our subgoal-guided policy $\pi^{\text{SG}}$ on both PHS* and LevinTS, which we denote PHS*($\pi^{\text{SG}}$) and LevinTS($\pi^{\text{SG}}$), respectively. Both PHS* and LevinTS were shown to perform well in single-agent deterministic search problems (Orseau & Lelis, 2021).

**Baselines.** To evaluate the training sample efficiency our subgoal-guided policy and data generation method provides from the trees when search terminates prematurely, we compare PHS*($\pi^{\text{SG}}$) to PHS* and LevinTS($\pi^{\text{SG}}$) to LevinTS using their original single policy formulation, denoted PHS*($\pi$) and LevinTS($\pi$), respectively. Following Orseau & Lelis (2021), we also evaluate our policy formulation against Weighted A* (Pohl, 1970) with $w = 1.5$, which was shown to be more sample efficient than PHS* in some environments. We also test our method against HIPS-$\varepsilon$, which is another complete subgoal-guided tree search method.

---

[1]https://github.com/jacobandreas/psketch/tree/master

## 4.3. Experimentation Procedure

For evaluating the sample efficiency of different methods to learn an effective policy, we use the setup of Orseau & Lelis (2021). A loss of $\ell(\cdot) = 1$ is used, which corresponds to the search $L(S, n_k^*)$ being the number of node expansions that an algorithm $S$ requires to solve problem instance $k$. PHS*$(\pi)$, LevinTS$(\pi)$, and WA* are trained using the Bootstrap process (Section 2.1 and Appendix C). PHS*$(\pi^{SG})$ and LevinTS$(\pi^{SG})$ are trained using the process outlined in Section 3 (with full pseudocode given in Appendix F). The Bootstrap process starts with a budget of 4000 expansions and use a budgeting update schedule noted in Appendix F. During the online training Bootstrap process, we record the number of outstanding training problems which have yet to be solved, and the total accumulated loss (expansions) incurred after every iteration.

The training procedure outlined for HIPS-$\varepsilon$ requires a pre-computed dataset of solution trajectories. Since it is not clear how to train HIPS-$\varepsilon$ online using the Bootstrap process, we do not include HIPS-$\varepsilon$ in our training efficiency results. The policy, heuristic, and subgoal generator models used in HIPS-$\varepsilon$ are trained following the procedure of Kujanpää et al. (2024). For Sokoban, we use the offline datasets provided by Kujanpää et al. (2024). For Box-World, CraftWorld, and Boulderdash, we use trajectories found by domain-specific solvers over the same training set used to train the other systems with the Bootstrap process.

Every domain has a disjoint set of 10,000 problem instances to train, 1,000 as validation, and 100 in the test set. Each algorithm is trained on 5 separate seeds, which determine the model initialization, the train and validate problem instances partitions, and how the problems are shuffled when batched. For the training efficiency experiment, we report the maximum, minimum, and average number of outstanding problems in our plot of results. For testing, we record the number of problem instances each algorithm was able to solve, the average number of expansions required to solve those instances, and the average solution length. We report the version of the trained network which allowed it to solve the highest number of problems, with tie breaks going to the one whose total solution lengths were the shortest (Orseau & Lelis, 2021). Due to the large runtime costs of HIPS-$\varepsilon$, we limit the budget during testing to 8,000.

## 4.4. Results

The first set of training loss experiments focuses on domains that all the baseline methods can solve. The total search loss incurred over the course of training, as measured by the number of node expansions, is shown in Figure 2.a. In the Box-World, CraftWorld, and BoulderDash environments, the subgoal-guided policy formulation PHS*$(\pi^{SG})$ and LevinTS$(\pi^{SG})$ require substantially fewer node expan-

Table 1: Results on the hard environments. "Expansions" represents the sum over all problems, time is measured in hours, and the maximum value for "Solved" is 10,000.

| ALGORITHM | EXPANSIONS | TIME | SOLVED |
|---|---|---|---|
| CRAFTWORLD (HARD) | | | |
| WA* (1.5) | 350,169,632 | 16.01 | 1,850 |
| LEVINTS$(\pi)$ | 360,454,214 | 16.02 | 73 |
| PHS*$(\pi)$ | 324,718,860 | 16.04 | 54 |
| LEVINTS$(\pi^{SG})$ | 172,188,794 | 16.02 | 9 |
| PHS*$(\pi^{SG})$ | **125,089,420** | **13.74** | **9,985** |
| BOULDERDASH (HARD) | | | |
| WA* (1.5) | 599,818,595 | 16.01 | 18 |
| LEVINTS$(\pi)$ | 636,718,638 | 16.04 | 16 |
| PHS*$(\pi)$ | 591,135,427 | 16.02 | 19 |
| LEVINTS$(\pi^{SG})$ | 284,044,919 | 16.02 | 31 |
| PHS*$(\pi^{SG})$ | **85,470,564** | **6.25** | **10,000** |
| TSP (HARD) | | | |
| WA* (1.5) | 1,019,840,000 | 16.01 | 0 |
| LEVINTS$(\pi)$ | 1,107,904,000 | 16.02 | 0 |
| PHS*$(\pi)$ | 980,928,000 | 16.02 | 0 |
| LEVINTS$(\pi^{SG})$ | 105,838,613 | 4.55 | **10,000** |
| PHS*$(\pi^{SG})$ | **20,593,798** | **1.23** | **10,000** |

sions than PHS*$(\pi)$ and LevinTS$(\pi)$, respectively, to solve all 10,000 training instances. Sokoban is the only environment where PHS*$(\pi)$ initially can solve more problems using fewer expansions, but both PHS*$(\pi)$ and PHS*$(\pi^{SG})$ converge towards the end of training when only the harder problems remain to be solved. We hypothesize that our method performs worse in Sokoban than in the other problems due to the training data that the clustering algorithm generates. While clustering finds important structures in the other problems, it fails to find helpful structures in Sokoban.

The next set of experiments is similar to the above, but instead, it showcases how the sample efficiency of our method enables policies to be learned in more challenging problems. For the domain environments with access to level-generators, problem instances were generated to be sufficiently hard, such as adding more diamonds for the agent to collect in BoulderDash. All methods were given a maximum of 16 hours for the online bootstrap training process, with the training loss curves given in Figure 2.b and summarized in Table 1. The baseline algorithms were not able to solve many of the training instances within the time budget, and incurred a large training loss. PHS*$(\pi^{SG})$ was able to complete the training process well under the time allocated on all problems evaluated. Despite our method requiring a higher computational cost per node expansion due to the use of several networks, its sample efficiency allows it to solve challenging problems while incurring a substantially smaller cost in terms of both node expansions and computation time.

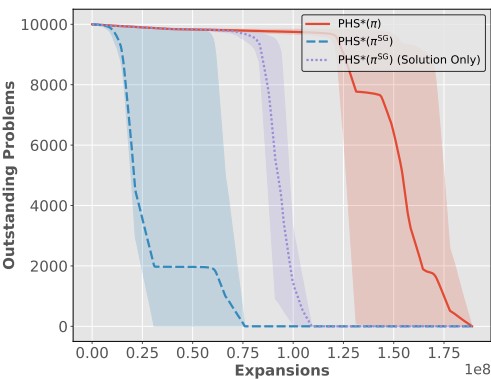

Figure 3: Reduction in search loss during training when using data from non-solution trees. The line represents the average and the shaded regions show maximum, minimum, and average outstanding problems across all seeds.

The results on the easier and more difficult domains support our hypothesis that our subgoal-guided policies can improve the sampling efficiency of policy-guided search algorithms.

**Search Loss Efficiency from Non-Solution Trees**.

One of the main contributions of our method for learning policies is that it can use data from failed searches to train its low-level policy and subgoal generator. While the previous experiment evaluated the training efficiency that the combination of subgoal-guided policy and the training method provides, this experiment aims to analyze the contribution of each. In Figure 3, we show the search loss during training for PHS*($\pi^{SG}$) in the BoulderDash environment when we limit it to only using solution trajectories to update the models. These results show that learning $\pi^{SG}$ from solution trajectories already substantially improves the sample efficiency of PHS*. We see another improvement when the system learns $\pi^{SG}$ from both successful and failed searches.

**Test Results**. To test the quality of the learned policies, Table 2 reports the average number of expansions and average solution length of each algorithm on the test set of the standard problems. On all domains except for Sokoban, PHS*($\pi^{SG}$) solved the test problems using the least number of expansions on average. WA* is able to find solutions that are the closest on average to the optimal cost, but it requires considerably more node expansions. With the exception of BoulderDash, LevinTS($\pi^{SG}$) and PHS*($\pi^{SG}$) can solve the test problems using fewer node expansions than LevinTS($\pi$) and PHS*($\pi$), respectively. Taken together, Table 2 and Figure 2 show that learning $\pi^{SG}$ can be more sample efficient than learning non-subgoal-based policies (Figure 2), while asymptotically producing policies of similar quality in terms of expansions and solution cost (Table 2).

HIPS-$\varepsilon$ was unable to solve any problem in BoulderDash, and the majority of problems in CraftWorld within budget

Table 2: Results on the test set for each of the standard domain environments. Expansions and solution length are averaged over solved problems. (†) Expansions reported for HIPS-$\varepsilon$ are only for the latent-level space, and thus are not directly comparable to the other algorithms.

| Algorithm | Solved | Expansions | Length |
|---|---|---|---|
| **Box-World (Standard)** | | | |
| WA* (1.5) | 100 | 2,398.68 | **66.19** |
| LevinTS($\pi$) | 100 | 605.05 | 67.91 |
| PHS*($\pi$) | 100 | 767.06 | 68.07 |
| HIPS-$\varepsilon$ | 100 | 467.79(†) | 67.39 |
| LevinTS($\pi^{SG}$) | 100 | 411.38 | 66.73 |
| PHS*($\pi^{SG}$) | 100 | **378.58** | 66.75 |
| **CraftWorld (Standard)** | | | |
| WA* (1.5) | 100 | 2,318.22 | **90.01** |
| LevinTS($\pi$) | 100 | 262.76 | 93.93 |
| PHS*($\pi$) | 100 | 172.04 | 93.49 |
| HIPS-$\varepsilon$ | 19 | 116.11(†) | 92.47 |
| LevinTS($\pi^{SG}$) | 100 | 208.28 | 93.93 |
| PHS*($\pi^{SG}$) | 100 | **103.23** | 94.54 |
| **BoulderDash (Standard)** | | | |
| WA* (1.5) | 100 | 1,193.60 | **51.44** |
| LevinTS($\pi$) | 100 | 61.33 | 52.90 |
| PHS*($\pi$) | 100 | 53.65 | 52.74 |
| HIPS-$\varepsilon$ | 0 | — | — |
| LevinTS($\pi^{SG}$) | 100 | 65.48 | 53.30 |
| PHS*($\pi^{SG}$) | 100 | **53.34** | 52.68 |
| **Sokoban (Standard)** | | | |
| WA* (1.5) | 100 | 1,091.45 | **32.81** |
| LevinTS($\pi$) | 100 | 1,177.26 | 41.04 |
| PHS*($\pi$) | 100 | 1,523.38 | 39.40 |
| HIPS-$\varepsilon$ | 100 | **80.55**(†) | 45.24 |
| LevinTS($\pi^{SG}$) | 100 | 496.87 | 41.85 |
| PHS*($\pi^{SG}$) | 100 | 808.42 | 39.61 |

constraints. HIPS-$\varepsilon$ performs poorly in these domains because its search is performed at the learned subgoal level. In domains whose observations are complex to represent, such as BoulderDash and CraftWorld, HIPS-$\varepsilon$ requires the VQVAE subgoal generator to be very accurate and not miss important features, such as the agent, goal, or key object locations. In environments where the VQVAE can reliably reconstruct the states, such as Sokoban, HIPS-$\varepsilon$ performed well and solved all problem instances. Note that expansions being reported by HIPS-$\varepsilon$ are not directly comparable to the other algorithms, as it only counts the expansions performed in the learned subgoal space, which does not include the costs such as grounding and verifying the actual paths in the original state space. Since PHS*($\pi^{SG}$) and LevinTS($\pi^{SG}$) perform the search in the original space, it is more forgiving with reconstruction errors from the subgoal generator.

Table 3 reports similar metrics for each algorithm on the test set of the hard problems. Each problem instance was given a

Table 3: Results on the test set for each of the hard domain environments. Expansions and solution length are averaged over solved problems.

| Algorithm | Solved | Expansions | Length |
|---|---|---|---|
| CraftWorld (Hard) | | | |
| WA* (1.5) | 100 | 313,572.52 | **117.03** |
| LevinTS($\pi$) | 0 | — | — |
| PHS*($\pi$) | 0 | — | — |
| LevinTS($\pi^{SG}$) | 100 | 395,557.94 | 123.29 |
| PHS*($\pi^{SG}$) | 100 | **3,071.83** | 120.77 |
| BoulderDash (Hard) | | | |
| WA* (1.5) | 16 | 271,636.94 | 58.19 |
| LevinTS($\pi$) | 0 | — | — |
| PHS*($\pi$) | 0 | — | — |
| LevinTS($\pi^{SG}$) | 22 | 315,807.91 | 69.5 |
| PHS*($\pi^{SG}$) | 100 | **172.32** | **84.5** |
| TSP (Hard) | | | |
| WA* (1.5) | 1 | 502,624.0 | 45.0 |
| LevinTS($\pi$) | 0 | — | — |
| PHS*($\pi$) | 0 | — | — |
| LevinTS($\pi^{SG}$) | 100 | 570.09 | **40.89** |
| PHS*($\pi^{SG}$) | 100 | **41.93** | 41.73 |

maximum budget of 512,000 node expansions. PHS*($\pi^{SG}$) was able to solve all test problems, while the other comparison methods struggled to solve a majority of them. The exception is WA* on CraftWorld, but it required over 100 times more node expansions as compared to PHS*($\pi^{SG}$). Due to the complexity of the problem instances, HIPS-$\varepsilon$ is omitted from these results as it requires a dataset of solution trajectories to train its policies and subgoal generator.

## 5. Related Work

**Subgoal Search**. Our work is primarily related to other works that use subgoals in search. kSubS (Czechowski et al., 2021) learns a subgoal generator to predict resulting states that are $k$ steps ahead of the current state. AdaSubS (Zawalski et al., 2022) extends kSubS by learning multiple subgoal generators, each for a different look-ahead depth $k$. Since both kSubS and AdaSubS search at the subgoal level, they are not complete. HIPS (Kujanpää et al., 2023) learns to segment trajectories through reinforcement learning, then trains a subgoal generator using these trajectories and a low-level policy to reach the generated subgoals. HIPS also searches at the subgoal level, and thus is not complete. HIPS-$\varepsilon$ (Kujanpää et al., 2024) augments the subgoal search by combining the subgoals and the actions of the environment. These approaches require a precomputed dataset of solution trajectories, which our approach does not require. Also, we showed empirically that HIPS-$\varepsilon$ might fail to solve problems when reconstructing the state is challenging, while our approach is robust to reconstruction inaccuracies.

**Goal-Conditioned Reinforcement Learning**: Identifying useful subgoal states for policies to navigate between has shown promising results for solving long-horizon tasks. There are many different approaches to how these subgoal states are generated from the history of states the agent has visited. HIGL (Kim et al., 2021) builds a graph of landmark states, from which a goal is selected. HILL (Zhang et al., 2023) and HESS (Li et al., 2021) sample goal states from a learned embedding using distance metrics. $L^3P$ (Zhang et al., 2021) learns a latent space and then learns latent landmarks through clustering, and finally decodes the cluster centroids as goals. DisTop (Aubret et al., 2023) clusters an embedding of the state space and samples a goal from one of the selected clusters. Our method is different in that it subgoals are found using only the underlying properties of the state-space graph, without relying on additional metrics to select subgoals such as novelty and reachability measures.

**VQVAEs as Subgoal Generators**: Vector Quantized Variational Autoencoders (Van Den Oord et al., 2017) have become a popular model to use as subgoal generator. DGRL (Islam et al., 2022), Choreographer (Mazzaglia et al., 2022), and QPHIL (Canesse et al., 2024) use VQVAEs to generate a target subgoal observation, which is used with the current state as input to policies. VQVAEs are generally trained on an offline dataset, separate from the reinforcement learning task. CQM (Lee et al., 2023) trains a VQVAE to discretize continuous observations, which is used to create a curriculum over landmarks. Our method trains a VQVAE to generate subgoal observations, but does so online, while learning the policy. This allows our method to be used in scenarios where generating quality training trajectories is too costly.

## 6. Conclusion

In this paper, we presented a novel way to combine subgoal discovery with subgoal-guided policies, which can be used as a drop-in replacement to policies used in existing policy tree search algorithms, such as LevinTS and PHS*. The neural network models in our method can be trained online without needing an offline solution dataset. Moreover, it can be trained from both successful and failed searches, which we showed to improve the sampling efficiency of tree search algorithms substantially. This is due to the fact that a subgoal conditioned policy can be trained using non-solution trajectories from the search tree. In the domains evaluated, our experiments showed that the large reduction in environment interactions does not reduce the quality of the solutions found, while requiring fewer expansions at test time. As we increased the problem's difficulty, the gap between our subgoal-based approach and other approaches increased—while our learned policies solved all test problems, previous approaches failed to learn effective ones.

## Acknowledgements

This research was supported by Canada's NSERC and the CIFAR AI Chairs program. This research was enabled in part by support provided by the Digital Research Alliance of Canada. The authors thank the anonymous reviewers for valuable feedback on this work.

## Impact Statement

This paper advances the field of Machine Learning by demonstrating how to improve existing tree search approaches to solve difficult "needle-in-the-haystack" single-agent problems. While the primary impact of our work is technical and our evaluated domains are game-like, this research direction could have broader applications in the long run. We do not see immediate societal risks, but as with any research in this field, ethical considerations should be taken into account when deploying these methods in real-world settings.

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

# A. Best-First Search

**Algorithm 1** Best-First Search (BFS)

1:  **Input:** Initial state $s_0$, evaluation function $\varphi$, budget $B$
2:  **Output:** Solution node, or status of failed search
3:  `struct Node {state s, float g}`
4:  $q \leftarrow$ `priority_queue(order_by = $\varphi$)`
5:  $v \leftarrow$ `set()`                                                                  {Set of visited states.}
6:  $n_0 =$ `Node`$(s_0, 0)$                                                          {Root node.}
7:  `insert`$(q, n_0, \varphi(n_0))$
8:  $b \leftarrow 0$
9:  **while** $q$ not empty **and** $b < B$ **do**
10:     $n \leftarrow$ `extract_min`$(q)$
11:     $s \leftarrow$ `state`$(n)$
12:     $b \leftarrow b + 1$
13:     **if** $s \in v$ **then**
14:       **continue**                                                                {Previously expanded state.}
15:     **end if**
16:     `insert`$(v, s)$
17:     **for each** $n' \in$ `expand`$(n)$ **do**
18:       $s' \leftarrow$ `state`$(n')$
19:       **if** `is_goal`$(s')$ **then**
20:          **return** $n'$
21:       **end if**
22:       `insert`$(q, n', \varphi(n'))$
23:     **end for**
24: **end while**
25: **return** `timeout` **if** $b \geq B$ **else** `no_solution`

# B. Policy Tree Search

Algorithm 2 and Algorithm 3 depict the policy tree search algorithms Levin Tree Search (LevinTS) and Policy-Guided Heuristic Search (PHS*) respectively.

**Algorithm 2** Levin Tree Search (LevinTS)

1:  **Input:** Initial state $s_0$, budget $b$
2:  **Output:** Solution node, or status of failed search
3:  **return** `BFS`$(s_0, \varphi_{\text{LevinTS}}, b)$

**Algorithm 3** Policy-Guided Heuristic Tree Search (PHS*)

1:  **Input:** Initial state $s_0$, budget $b$
2:  **Output:** Solution node, or status of failed search
3:  **return** `BFS`$(s_0, \varphi_{\text{PHS}}, b)$

## C. The Bootstrap Training Process

Algorithm 4 depicts the Bootstrap Process. After each iteration of the algorithm, a new budget needs to be selected. Arfaee et al. (2011) unconditionally double the budget every iteration. Orseau & Lelis (2021) double the budget only if no new problems are solved in the current iteration.

---

**Algorithm 4** The Bootstrap Process

---

1: **Input:** Initial budget $B_0$, root states $\mathcal{S}^0$ (training problems), search algorithm $S$, neural policy/heuristic models $\Theta$
2: `solved` $\leftarrow$ `set()`
3: **for** $t = 0, 1, \ldots$ **do**
4:    **for each** $s_0 \in \mathcal{S}^0$ **do**
5:       `result` $\leftarrow S(s_0, \Theta, B_t)$
6:       **if** `result` $\neq$ `timeout` **and** `result` $\neq$ `no_solution` **then**
7:          `insert(solved,` $n_0$`)`
8:          update $\Theta$ using solution trajectory from `result`
9:       **end if**
10:    **end for**
11:    **if** $|\text{solved}| = |\mathcal{N}^0|$ **then**
12:       **break**
13:    **end if**
14:    `choose budget` $B_{t+1}$
15: **end for**

---

## D. Vector Quantized Variational Autoencoders

VQVAEs utilize a codebook containing $k$ trainable $D$-dimensional embedding vectors $\boldsymbol{e}_i \in \mathbb{R}^D, i = 1, \ldots, k$. Inputs $\boldsymbol{x}$ are encoded by the encoder, resulting in an encoded vector $\boldsymbol{z}_e$. The encoding vector is then mapped to a quantized encoding vector $\boldsymbol{z}_q$ which is the *nearest* embedding vector using Euclidean distance:

$$\boldsymbol{z}_q = \boldsymbol{e}_c, \qquad \text{where } c = \arg\min_i \|\boldsymbol{z}_e - \boldsymbol{e}_i\|_2. \tag{5}$$

The discretized encoding vector is then passed through the decoder which produces a reconstructed output $\hat{\boldsymbol{x}}$. The loss function used to optimize the encoder, decoder, and codebook is given by

$$\mathcal{L}_{VQ} = \mathcal{L}_{\text{rec}}(\hat{\boldsymbol{x}}, \boldsymbol{x}) + \|sg(\boldsymbol{z}_e) - \boldsymbol{e}_i\|_2^2 + \beta \|\boldsymbol{z}_e - sg(\boldsymbol{e}_i)\|_2^2 \tag{6}$$

where $\mathcal{L}_{\text{rec}}$ is the reconstruction loss, and $sg(\cdot)$ is the stop gradient operation. The stop gradient is required so that we can independently update the codebook to converge to the encoding vector, and the encoding vector to converge to the codebook.

# E. Louvain Algorithm

A *partition* of a graph is a grouping of its nodes into mutually exclusive groups called *clusters*. The *modularity* (Newman & Girvan, 2004; Leicht & Newman, 2008) of a graph measures the relative density of edges inside the clusters to outgoing edges. The modularity of a cluster $Q_c$ is given by

$$Q_c = \frac{\Sigma_{\text{in}}^C}{2m} - \rho \left( \frac{\Sigma_{\text{tot}}^C}{2m} \right)^2, \tag{7}$$

where $\Sigma_{\text{in}}^C$ is the total edge weight between nodes in cluster $c$, $\Sigma_{\text{tot}}^C$ is the total edge weight within the cluster (which includes edges leading out to other clusters), $m$ is the sum of all edge weights in the graph, and $\rho > 0$ is a balancing parameter. The modularity of a graph $G$ is then given by the sum of all cluster modularities: $Q = \sum_c Q_c$. A clustering that maximizes the modularity of the graph is one that has dense connections between nodes in the same clusters, with sparse connections between each of the clusters.

It has been shown that finding a partition of a graph that maximizes the modularity is NP-hard (Brandes et al., 2006), so approximation algorithms are generally used. The Louvain algorithm (Blondel et al., 2008) is an iterative hierarchical graph clustering algorithm. It works by first placing every node of the given graph $G_0$ into its own cluster. Nodes are then moved into a neighbouring cluster iteratively if it increases the overall modularity of the graph. This process stops once no additional progress can be made. The result is a clustering over the graph, which can then be used to create a new aggregate graph $G_1$ as follows: clusters of $G_0$ become nodes in graph $G_1$, and two nodes in $G_1$ have an edge joining them if there exists an edge that connects neighbouring clusters in $G_0$. The resulting graph $G_1$ can then be used in this process again, which will produce another aggregate graph $G_2$, and so on until either the final graph has a single node or the returned graph $G_{i+1}$ is the same graph as $G_i$ (*i.e.*, no nodes were able to be moved to a different cluster). The result from this process is hierarchy of graph clusterings, in which earlier graphs contain many smaller clusters which are then merged into fewer larger clusters.

The Louvain algorithm is given as pseudocode in Algorithm 5, which is adapted from Evans & Şimşek (2023).

---

**Algorithm 5** The Louvain Algorithm

---

1: **Input:** Base graph $G_0 = (V_0, E_0)$
2: **Output:** Hierarchy of graph clusterings
3: **for** $i = 0, 1, \ldots$ **do**
4:    $C_i \leftarrow \{\{u\} \mid u \in V_i\}$                                                       {initialize singleton cluster over each node.}
5:    $Q_{\text{old}} \leftarrow \sum_i \left( \Sigma_{\text{in}}^{C_i}/2m - \rho(\Sigma_{\text{tot}}^{C_i})^2/(2m)^2 \right)$                              {Modularity of $C_i$, Equation 7.}
6:    **repeat**
7:       **for each** $u \in V_i$ **do**
8:          Find cluster $c \in C_i$ node $u$ belongs to
9:          Find neighbouring clusters $N_c$ of $c$
10:         Remove $u$ from cluster $c$
11:         Compute modularity gains by moving $u$ into each neighbouring cluster $N_c$
12:         Move $u$ into neighbouring cluster maximizing modularity, if gain $> 0$
13:       **end for**
14:    **until** no change in $C_i$
15:    $Q_{\text{new}} \leftarrow \sum_i \left( \Sigma_{\text{in}}^{C_i}/2m - \rho(\Sigma_{\text{tot}}^{C_i})^2/(2m)^2 \right)$                        {Modularity of updated $C_i$, Equation 7.}
16:    **if** $Q_{\text{new}} > Q_{\text{old}}$ **then**
17:       $V_{i+1} \leftarrow \{c \mid c \in C_i\}$                           {Parent graph $G_{i+1}$ where nodes are clusters from $G_i$.}
18:       $E_{i+1} \leftarrow \{(c_i, c_j) \mid c_i, c_j \in C_i, c_i \text{ and } c_j \text{ are neighbours in } C_i\}$
19:       $G_{i+1} \leftarrow (V_{i+1}, E_{i+1})$
20:    **else**
21:       **break**                                                                       {No further gain.}
22:    **end if**
23: **end for**
24: **return** $(G_0, \ldots, G_N)$

---

## F. Complete Training Procedure

Algorithm 6 depicts our modified Bootstrap training algorithm which we use for PHS*($\pi^{\text{SG}}$) and LevinTS($\pi^{\text{SG}}$). After each iteration of the algorithm, a new budget needs to be selected. In our experiments, we follow a budget selection method which follows Orseau et al. (2023): If more than a factor $(1 + b)$ of problems are solved in the current iteration $t$ than was at the previous iteration $t - 1$, the next budget is reduced to $B_{t+1} = \max(B_0, B_t/2)$. Otherwise, the budget is increased to $B_{t+1} = 2B_t + T_t/s_t$, where $T_t$ is the total number of expansions used for the solved problems, and $s_t$ is the number of remaining unsolved problems.

---

**Algorithm 6** Subgoal-Guided Policy Search Training Procedure

---

1: **Input:** Initial budget $B_0$, root nodes $\mathcal{N}^0$ (training problems), low-level conditional policy $\pi_\theta^{\text{low}}$, high-level subgoal policy $\pi_\psi^{\text{hi}}$, heuristic model $h_\omega$ Subgoal Generator $VQ_\phi$, Louvain sampling level $k$, Policy Search algorithm $S$
2: $\texttt{solved} \leftarrow \texttt{set}()$
3: $\texttt{lengths} \leftarrow []$
4: **for** $t = 0, 1, \ldots$ **do**
5:   **for each** $n_0 \in \mathcal{N}^0$ **do**
6:     $\texttt{result} \leftarrow \texttt{S}(n_0, [\pi_\theta^{\text{low}}, \pi_\psi^{\text{hi}}, h_\omega, VQ_\phi], B_t)$
7:     **if** $\texttt{result} \neq \texttt{timeout}$ **and** $\texttt{result} \neq \texttt{no\_solution}$ **then**
8:       $\texttt{insert}(\texttt{solved}, n_0)$
9:       $T = (s_0, a_0, s_1, \ldots, s_t) \leftarrow$ reconstructed path from initial state $\texttt{state}(n_0)$ to goal state $\texttt{state}(n^*)$
10:       Optimize $h_\omega$ for each state $s_i \in T$ by minimizing the mean-squared error with respect to the distance to $s_t$.
11:       $T' \leftarrow$ partition $T$ into sub-trajectories of lengths sampled from a Normal distribution parameterized by the mean and variance of $\texttt{lengths}$
12:       **for each** $PT = (s_i, a_i, s_{i+1}, \ldots, s_{i+t'}) \in T'$ **do**
13:         $\hat{s}_{i+t'} \leftarrow VQ_\phi(s_i, s_{i+t'})$
14:         $\hat{s}_g \leftarrow \{\hat{s}_{g_i}\}_{i=1}^k = \{VQ_\phi(e_i, s_i)\}_{i=1}^k$
15:         $c \leftarrow$ quantized codebook index from $VQ_\phi(s_i, s_{i+t'})$
16:         Optimize $VQ_\phi$ by minimizing the loss $\mathcal{L}_{VQ}(\hat{s}_{i+t'}, s_{i+t'})$
17:         Optimize $\pi_\theta^{\text{low}}(a_j|s_j, s_{i+t'})$ for each $s_j, a_j \in PT$ by minimizing the cross-entropy loss with respect to $a_i$
18:         Optimize $\pi_\psi^{\text{hi}}(\hat{s}_g|s_j)$ for each $s_j \in PT$ by minimizing the cross-entropy loss with respect to $\hat{s}_{g_c}$
19:       **end for**
20:     **else**
21:       $G_0 \leftarrow$ underlying graph from search tree
22:       $G_k \leftarrow k$th cluster graph from $\texttt{Louvain}(G_0)$
23:       $C_i, C_j \leftarrow$ sampled neighbour clusters from $G_k$
24:       $s_{\text{cur}} \leftarrow$ sampled state from $C_i$
25:       $s_{\text{tar}} \leftarrow$ sampled state from $C_j$
26:       $T = (s_0, a_0, s_1, \ldots, s_t) \leftarrow$ reconstructed path from $s_{\text{cur}}$ to $s_{\text{tar}}$
27:       $\texttt{insert}(\texttt{lengths}, t)$
28:       $\hat{s}_{\text{tar}} \leftarrow VQ_\phi(s_{\text{cur}}, s_{\text{tar}})$
29:       Optimize $VQ_\phi$ by minimizing the loss $\mathcal{L}_{VQ}(\hat{s}_{\text{tar}}, s_{\text{tar}})$
30:       Optimize $\pi_\theta^{\text{low}}(a_i|s_i, s_{\text{tar}})$ for each $s_i, a_i \in T$ by minimizing the cross-entropy loss with respect to $a_i$
31:     **end if**
32:   **end for**
33:   **if** $|\texttt{solved}| = |\mathcal{N}^0|$ **then**
34:     **break**
35:   **end if**
36:   choose budget $B_{t+1}$
37: **end for**

---

## G. Implementation and Machine Details

The algorithms and environment are implemented in C++, adhering to the C++20 standard. The codebase [2] is compiled using the *GNU Compiler Collection* version 13.3.0, and uses the PyTorch 2.4 C++ frontend (Paszke et al., 2019). Where available, the official implementation of comparison methods are used. All experiments were conducted on an Intel i9-7960X and Nvidia 3090, with 128GB of system memory running Ubuntu 24.04.

## H. Additional Environment Details

Most of the problem instances used in this paper are based off of existing open source implementations. Box-World uses an open source problem generator [3] with a goal length of 5, and several distractor paths of length 3. A distractor path is a sequence of key and locks which can be used by the agent, but does not progress to the goal. CraftWorld uses a custom problem generator based off of (Andreas et al., 2017), which has various levels of difficulty for the generated problems. BoulderDash also uses a custom problem generator which randomly places coloured keys, doors, diamonds, and the exit in different rooms. Sokoban uses the *Boxban* [4] problems. Finally, TSP uses a custom level generator to randomly place the agent starting location and the cities.

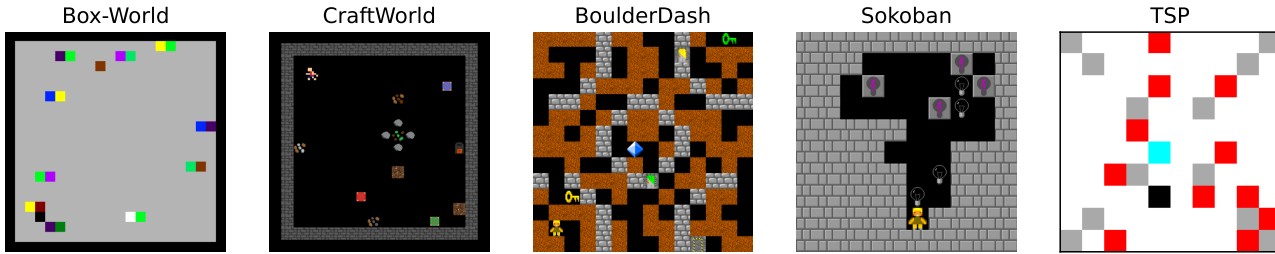

| Box-World | CraftWorld | BoulderDash | Sokoban | TSP |

Figure 4: The environment domains used in the experiments. Box-World: The agent in black needs to open colored locks (right pixel) to receive a colored key (left pixel) until it gets to the goal (white). CraftWorld: The agent must create an iron pickaxe to get the gem so that they can craft a ring. BoulderDash: The agent must get the green key to unlock the room containing the diamond. Once the diamond is collect, the exit in the bottom-right room will open. Sokoban: The agent must push boxes to the goal locations without getting boxes stuck in corners. TSP: The agent (black) must visit each city (red) then return back to the first city (blue). The gray boxes are traversable obstacles.

## I. Additional Experimental Details

In our experiments, PHS*($\pi^{\text{SG}}$), LevinTS($\pi^{\text{SG}}$), PHS*($\pi$), LevinTS($\pi$), and WA* all use a ResNet-based (He et al., 2016) networks. PHS*($\pi$) uses a single network with two heads, one for the policy and one for the heuristic. For PHS*($\pi^{\text{SG}}$), we use separate networks for the conditional low policy, and a two-headed network for the subgoal policy and global heuristic. LevinTS($\pi^{\text{SG}}$) follows the same network structure, without the heuristic head. We use the Adam optimizer (Kingma, 2014), with learning rate of 3E-4 and L2-regularization of 1E-4. The policy and heuristic networks for PHS*($\pi$), LevinTS($\pi$), PHS*($\pi^{\text{SG}}$), and LevinTS($\pi^{\text{SG}}$) both use 128 ResNet channels, with PHS*($\pi^{\text{SG}}$) and LevinTS($\pi^{\text{SG}}$) using half the number of blocks (4 versus 8) due to the fact that they both have both a low-level and high-level policy. The VQVAE subgoal generator uses a codebook size of 4, a codebook dimension of size 128, and $\beta = 0.25$. We use the open source implementation for HIPS-$\varepsilon$ in our experiments[5]. We use $\epsilon = $ 1E-3 for all environments, and use for version of HIPS-$\varepsilon$ which assumes access to an environment dynamics model. All other hyperparameter follows the authors choice in their Box-World experiments (Kujanpää et al., 2024).

---

[2] https://github.com/tuero/subgoal-guided-policy-search
[3] https://github.com/nathangrinsztajn/Box-World
[4] https://github.com/deepmind/boxoban-levels/
[5] https://github.com/kallekku/HIPS

## J. Ablation: Impact of Codebook Size

The size of the codebook in the VQVAE determines the number of how many subgoals are parameterized by the model. In Figure 5, we show the search loss during training for various codebook sizes on the CraftWorld environment. The chosen codebook size of $k = 4$ incurs the smallest search loss, but we note that there is a fair amount of overlap between the curves. Table 4 shows how each of these models performs on the test set. While the solution length is relatively the same, there is quite a large difference in the number of expansions required to solve the problems. If too few subgoals are used, then the codebook becomes overburdened in trying to represent a single subgoal state. Interestingly, if too many subgoals are used, we also see an increase in the number of expansions required. One possible reason for this could be that if a large number of codebook entries are not being used, then the resulting mixture policy will look more like a uniform policy, which will increase the number of node expansions required. One avenue for future work is to dynamically remove unused codebook entries from being used in the final policy.

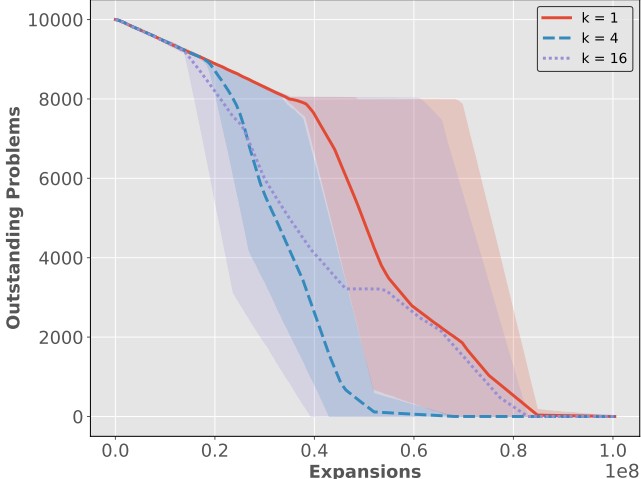

Figure 5: Learning curves during training using the Bootstrap process for varying codebook sizes. The line is the average outstanding problems for the respective number of expansions accumulated during training. All training runs lie within the shaded regions.

Table 4: Results on CraftWorld test set using various codebook sizes. Expansions and solution length are averaged over solved problems.

| ALGORITHM | SOLVED | EXPANSIONS | LENGTH |
|---|---|---|---|
| K = 1 | 100 | 263.92 | 92.52 |
| K = 4 | 100 | 103.23 | 94.54 |
| K = 16 | 100 | 305.17 | 94.16 |

## K. Ablation: Impact of Cluster Graph Level when Sampling Subgoals

From the base graph $G_0$, the Louvain algorithm produces a hierarchy of cluster graphs $\{G_1, \ldots, G_N\}$ where earlier graphs contain many smaller clusters with later graphs merging them into fewer clusters. One decision to make when sampling subgoals is which of these cluster graphs in the hierarchy to choose from. Smaller abstraction levels will generally result in pairs of states being sampled which are close in space/time, and larger abstraction levels will generally result in pairs of states further apart, which is depicted in Table 5.

Table 5: Average trajectory length between subgoals for each cluster level on the CraftWorld environment.

| CLUSTER GRAPH LEVEL | SUBGOAL DISTANCE |
|:---:|---:|
| 1 | 1.66 |
| 3 | 4.93 |
| 5 | 17.29 |

Figure 6 shows the search loss during training when sampling subgoals from various cluster graph levels, and Table 6 shows the test results on the CraftWorld environment. From the search loss during training and at test time, it appears that one must be careful in that they do not sample subgoals which are too close together from $G_1$, as the subgoals will not be as meaningful. Sampling subgoals from $G_5$ which were far apart as compared to $G_3$ did lead to a slight loss in performance, both in terms of number of node expansions and solution length.

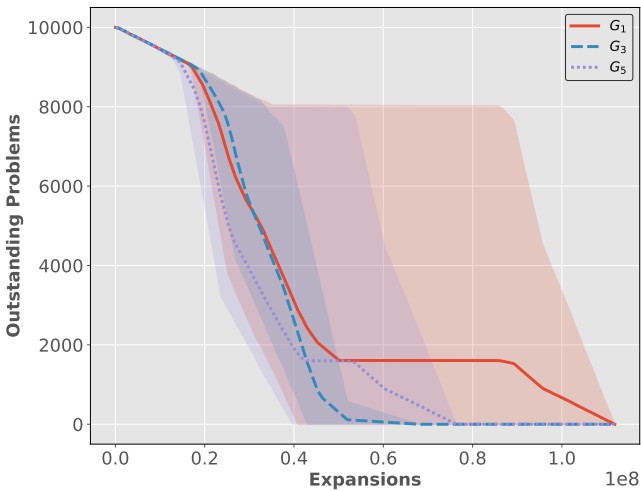

Figure 6: Learning curves during training using the Bootstrap process for varying cluster graph levels from which subgoal states were sampled from. The line is the average outstanding problems for the respective number of expansions accumulated during training. All training runs lie within the shaded regions.

Table 6: Results on CraftWorld test set using models which were trained using various cluster graph levels. Expansions and solution length are averaged over solved problems.

| ALGORITHM | SOLVED | EXPANSIONS | LENGTH |
|:---|:---:|:---:|:---:|
| G1 | 100.00 | 287.24 | 93.88 |
| G3 | 100.00 | 103.23 | 94.54 |
| G5 | 100.00 | 110.10 | 95.05 |

## L. Robustness to Out-of-Distribution Scenarios

Table 7 represents the results of an experiment that evaluates the algorithms on BoulderDash problems which require two keys to unlock two consecutive doors before reaching the diamond, whereas the neural policy and heuristic models were trained on problems where only a single key was required. PHS*$(\pi)$ and PHS*$(\pi^{SG})$ perform similarly, using a number of expansions close to the solution length which means that the policies learned can almost deterministically solve these problems. LevinTS$(\pi^{SG})$ can handle out-of-distribution problems much better than LevinTS$(\pi)$ in terms of the number of expansions required, while finding solutions slightly longer. WA* could not solve all problems, possibly for having to rely only on the guidance of a heuristic function. The heuristic WA* uses was trained to estimate the cost-to-go in a smaller interval than the longer out-of-distribution problems. Note that HIPS-$\varepsilon$ is not present here as it was unable to solve any BoulderDash problems in the original test set, due to the subgoal generator having difficulty to reproduce the complex state representation.

Table 7: Results on an out-of-distribution test set for BoulderDash. Expansions and solution length are averaged over solved problems.

| ALGORITHM | SOLVED | EXPANSIONS | LENGTH |
|---|---|---|---|
| WA* (1.5) | 73 | 9,712.64 | 68.29 |
| LEVINTS$(\pi)$ | 100 | 131.41 | **70.15** |
| PHS*$(\pi)$ | 100 | 72.45 | 70.37 |
| LEVINTS$(\pi^{SG})$ | 100 | 72.30 | 72.09 |
| PHS*$(\pi^{SG})$ | 100 | **71.35** | 70.25 |

