# OpenReview forum: "Subgoal-Guided Policy Heuristic Search with Learned Subgoals"
_ICML.cc/2025/Conference — ICML 2025 poster_

### Official Review · Reviewer_yx2F · 2025-03-05

**Overall Recommendation:** 3

**Summary:**

The paper proposes a new approach to training policies for search. The authors leverage subgoal guidance and this way they show how to get the training signal even from unsolved episodes. By experiments in 4 environments, they show that the proposed approach improves sample efficiency of the training procedure.

## Update after rebuttal

Thank you for the answer. Although I'm still on the fence about this paper, I will raise my rating to 3. That said, I still encourage the authors to make the proposed adjustments to strengthen the paper.

> Indeed, if the problems are on the easier end, then our method will not offer gains in terms of running time. [...] These results will be added to the paper.

I cannot give credit for results that are not shown. If that is indeed the case, then please include evaluations on harder instances in the paper. Otherwise, the paper will have limited impact, even if published. The results show currently are somewhat misleading as they rely only on the expansions metric that is not equally handled by methods; taking into account other metrics, such as running time currently makes the advantage rather small.

> Why We Prefer to Not Use the Trick of Filtering Out Actions

No, I am proposing a very simple baseline: fix k=2, $\varepsilon=0$, i=0, and j=$\infty$. To be honest, I expect that running such an experiment would take less time than writing the justification for not running it. All you have to do is to add `action_probs = action_probs * (action_probs >= torch.topk(action_probs, k).values.min())` after `action_probs = pi(s)` in the implementation of the baseline.

> is orthogonal to our subgoal-based policies

Yes, I agree. However, this is not exactly about improving speed. The point is to understand why your approach is beneficial. One advantage is that using the high-level policy for additional weighting makes the policy more focused on actions that matter. The question is whether this is the main benefit? If you show that naively focusing the search on top actions is still inferior to your approach, it would significantly strengthen your results.

**Claims And Evidence:**

The main claim of the paper is that the proposed training pipeline improves sample efficiency of the training. More spcifically -- less node expansions are required until the search algorithm can solve all instances. I am mostly convinced with that. However, a few concerns remain:

1. A single expansion of LevinTS/PHS requires one policy call and 0-k heuristic calls. On the other hand, a single expansion of your approaches require k VQ-VAE calls, 2k policy calls, and 0-k heuristic calls. This is a considerable difference, especially given that the VQ-VAE model likely has considerably more parameters than policies. Please discuss the difference in expansion cost and make it explicit in the paper. What is the magnitude of that difference in practice?

2. Given that your method requires training 3 policy-like models instead of a single one for baselines, is it not the case that it trains faster simply because in practice it receives more training updates? Please discuss that.

3. The cost of expansion is highly dependent on the number of generated subgoals. However, I found no specification of such hyperparameters (except few in appendices J,K). It makes a huge difference whether you use 4 or 40 subgoals. Please discuss that.

4. While I understand the proposed method, I am missing an intuition why it works. The key step lies in extracting the training signal from non-solution data. The pipeline you proposed uses those to train subgoal generator and low-level policy. This data is sufficient to train a goal-reaching low-level policy. The subgoals learned this way are not targetted toward the solution, hence the generator learns to output _any_ valid subgoals. While it makes sense that it should considerably improve the sample efficiency of the subgoal search, I don't see immediate reasons for why we should expect it to outperform plain low-level search. As far as I understand, the main role of subgoals in your pipeline is that they provide additional lookahead-driven weighting for actions. What would happen if you would augment baselines by "simulating" this subgoal guidance (e.g. during expansion, for each action make a single few-step policy rollout, treat the final state as a subgoal and the product of policy predictions on the path as subgoal probabilities)? Would it be sufficient to close the gap between methods?

5. There is also another difference: subgoals can be seen as a way to select more promising action from the full set. How would the performance of low-level methods change if you limited the number of actions used for each expansion to only top 2 according to the policy (or use weighting [0.45, 0.45, 0.05, 0.05] if you wish to preserve completeness)?

6. Were the key hyperparameters of each method tuned separately for each domain? From my experience, the weight of 1.5 for A* seems quite high, did you check other values as well?

I'm happy to accept the paper if my concerns are addressed.

**Essential References Not Discussed:**

I'm not aware of any essential references that have to be added.

**Experimental Designs Or Analyses:**

Yes, I checked all experiments in the main text.

**Methods And Evaluation Criteria:**

Yes.

**Other Comments Or Suggestions:**

- The running title is missing.
- typos: (90L "subgoals"), (74R "represent"), (238L "a hierarchy"), (251R "using")
- Figure 1.c.ii: there is a problem with indexing, all sequences have the same indices.
- Why using the Louvain algorithm is helpful? Is it much better than sampling subgoal pairs from unsolved trajectories in the same way as from solved trajectories?

**Other Strengths And Weaknesses:**

While the proposed pipeline mostly rely on existing parts, the proposed combination is novel and as such can be a good contribution.

**Questions For Authors:**

Please answer my questions from the Claims and Evidence section.

**Relation To Broader Scientific Literature:**

The proposed approach is mostly a combination of existing tools (LevinTS/PHS search, HIPS-inspired subgoal model, Louvain algorithm for subgoal discovery), but the proposed combination and additional training details are novel to my best knowledge.

**Theoretical Claims:**

There are no theoretical claims that need to be checked.

---

> ### Author Rebuttal · Authors · 2025-04-01
>
> Thank you for your helpful comments and feedback.
>
> **Concern 1**
>
> Each node generation for LevinTS/PHS requires a single policy call (plus heuristic for PHS). Each node generation for our approach requires K VQ-VAE calls to generate the K subgoal observations, K policy calls for the low-level policy (over each of the K generated subgoal target observations), and a single policy call for the high-level policy mixture. We make the K calls faster by batching them. Due to batching, the slowdown is not linear with the size of K, (see **Concern 3**).
>
> Consider the results of the following experiment on a much more difficult version of BoulderDash, following the same procedure from **Section 4.3**. We also increased the network size used in PHS*($\pi$) (denoted LARGE) to match the number of parameters used in PHS*($\pi^{SG}$) to show that simply using more parameters does not help.
>
> PHS*($\pi^{SG}$) learned how to solve the difficult set of BoulderDash within 6.25 hours. The baselines did not learn to solve the problems even when granted 4x more expansions and 2x more time. We also report the number of nodes generated per second of each model. We only have a factor of 2 slowdown while using 4 times more parameters and K subgoals.
>
> |Algorithm | Total Loss (Expansions) | Total Time (Hours) | Training Instances Solved | Nodes Generated per Second |
> |---|---:|---:|---:|--:|
> | WA* | 606,000,000 | 15.58 | 16 |1,872.31 |
> | LevinTS($\pi$) | 636,718,638 | 16.69 | 16 | 1,828.51 |
> | PHS*($\pi$) | 599,849,865 | 15.61 | 16 | 1,765.00 |
> | PHS*($\pi$) LARGE | 344,512,985 | 33.04 | 1 | 496.45 |
> | PHS*($\pi^{SG}$) [Ours] | 85,470,564 | 6.25 | 10,000 | 791.69 |
>
> We then tested the resulting models on a separate test set of 100 problems to match the procedure done for Table 1.
>
> | Algorithm | Solved | Expansions |
> | --- | --: | --: |
> | WA* | 14 | 272,872.28 |
> | LevinTS($\pi$) | 14 | 280,407.36 |
> | PHS*($\pi$) | 14 | 288,045.64 |
> | PHS*($\pi$) LARGE | 4 | 110,805.75 |
> | PHS*($\pi^{SG}$) [Ours] | 100 | 1,291.37 |
>
> **Concern 2**
>
> Yes, we equip the learning agent with the ability of learning from experiences that previous methods could not use for learning. We update our method more often because we learn how to reach the subgoals the clustering algorithm generates. The baselines cannot learn from such experience because they do not learn and exploit subgoals.
>
> **Concern 3**
>
> We make a single call of batch size K for a faster query. The time information was indeed missing in our ablation, which we include below. Due to batching, the number of generations per second does not decrease linearly with the number of subgoals.
>
> | Codebook Size (Number of Subgoals) | Nodes Generated per Second |
> | --- | ---: |
> | 1 | 266.85 |
> | 4 | 261.74 |
> | 16 | 208.44 |
>
> **Concern 4**
>
> A key difference between our approach and the baselines is that we can learn from problems we have yet to solve. As an example from BoulderDash, if many diamonds are required to solve a problem, it may be difficult to find a solution, but sampled paths may contain the agent collecting a single diamond. The low-level policy learns to collect diamonds even before the method can solve any problems (see Figure 3 of the paper).
>
> While we find the suggestion of performing roll-outs to simulate subgoal guidance interesting, we don’t see how it would be competitive with our method. Our method learns helpful subgoals because it uses the information contained in the CLOSED list of the search, which stores the entire search tree. The roll-out approach contains a single path. While the CLOSED list contains helpful information such as how to collect a key and open a door, it is unlikely a roll-out will capture this type of information.
>
> **Concern 5**
>
> We could apply the suggestion of using the top-k (e.g., k = 2) actions during search with the baselines in two moments: (1) during training and (2) during testing.
> - **Training**: Once the neural model is randomly initialized, the policy is nearly uniform. So choosing the top-k would be an arbitrary choice, as we would choose the two actions that happen to have a slightly higher probability at initialization.
> - **Testing**: Once these models manage to solve all training instances, their policies almost deterministically choose their actions. So choosing k > 1 would increase the number of expansions because we would be forcing the algorithm to explore when it can almost deterministically solve the problem.
>
> **Concern 6**
>
> We did not tune hyperparameters specifically for each domain for our method. For any
> hyperparameter specific to HIPS-e, we used their listed ones. For WA*, we followed LevinTS and PHS* which used a weight of 1.5. WA* tends to be quite robust to the weight value as it learns its heuristic function.

---

> > ### Comment · Reviewer_yx2F · 2025-04-02
> >
> > I acknowledge the answers. However, I'm still somewhat concerned about using the number of expansions as the main budget metric. I acknowledge that with batching the running time can be made ~2x times slower than baselines (actually the baselines can be batched as well, but with a bit more effort). However, if we account main results even for that number, the difference between $\pi^{SG}$ and baselines seem to disappear.
> >
> > Ad concern 5: As the training progresses, the models get better. At that point, it makes sense to expand only a limited number of best actions according to the policy estimates, not to waste the budget on worse moves. From my experience, such trick usually considerably improves efficiency, is easy to add, and is in line with how the proposed method works, so makes the comparison more fair.
> >
> > If the search action selection is not important neither during training nor testing, why search is used at all?

---

> > > ### Author Response · Authors · 2025-04-07
> > >
> > > Thank you for taking the time to respond. We appreciate the opportunity to address your concerns.
> > >
> > >
> > > **The Trick of Filtering Out Actions Can be Used with Our Approach**
> > >
> > > We failed to clarify in our initial response that the trick of filtering out actions the reviewer suggested is orthogonal to our subgoal-based policies and therefore can potentially be used to speed up the search of our approach too.
> > >
> > > The filtering out trick the reviewer suggests will increase the probability of the top $k$ actions and reduce the probability of the remaining actions. What the high-level policy $\pi^\text{hi}$ does is to increase/decrease the weight of a low-level policy and not of an action, as the trick does. Instead of increasing the probability values to a predefined value, our approach relies on an optimization process that minimizes the Levin loss—an upper bound on the size of the tree. As a result, the high-level policy $\pi^\text{hi}$ has its weights adjusted to minimize the size of the tree. This is in contrast with the more aggressive approach of setting the top $k$ actions to a predefined value.
> > >
> > > Once $\pi^\text{hi}$ is combined with the low-level policies as shown in Equation 4 of the paper, we obtain a probability distribution like any other. So there might be a point during training that our approach could also benefit from increasing the probability of the top $k$ actions, when the high and low-level policies are not as sharp as they will be at the end of training. We hope this clarifies this point, and it explains why our experiments are fair. They would be unfair if we only used the top-$k$ trick with the baselines and not with our approach.
> > >
> > > **Why We Prefer to Not Use the Trick of Filtering Out Actions**
> > >
> > > In our original response we explained that we cannot use the trick of picking $k$ actions and maximizing their probabilities while keeping the other actions at a minimum $\epsilon$ in the beginning and at the end of training. In the early stages of learning, the probability of all actions is similar and the decision of which actions to maximize will be arbitrary; at the later stages of learning the policy is nearly deterministic, so maximizing $k > 1$ actions will hamper performance. Can the trick work in between these two extremes? Yes, it can. However, you need to define the following to make it work:
> > > 1. Choose the number of actions $k$ to maximize.
> > > 2. Choose the value $\epsilon$.
> > > 3. Choose the learning iteration $i$ in which to start using the trick (we cannot use it in the beginning of training and we cannot use it at the end of training).
> > > 4. Choose the learning iteration $j > i$ in which to stop using the trick.
> > >
> > > If we sweep over different values of $k$, $\epsilon$, $i$, and $j$, we will possibly find values that will speed up learning. However, by the time we finish the sweep, the algorithms that do not use the trick will likely have finished training because they do not need to set any of these hyperparameters.
> > >
> > > Importantly, LevinTS and PHS* are trained to minimize the Levin loss, an upper bound on the number of nodes they expand to solve a problem. The training process we use is principled because we are learning to minimize the search tree of these algorithms. The trick of filtering out actions would make us lose this important property.
> > >
> > > Since the trick requires us to set a number of hyperparameters and we would lose the property that makes us excited about LevinTS and PHS, we prefer not to use it in our experiments.
> > >
> > > **Running Time**
> > >
> > > Indeed, if the problems are on the easier end, then our method will not offer gains in terms of running time. However, note that we showed in our initial response that, for harder problems, such as BoulderDash with more keys and diamonds, the difference between our method and the baselines is so large that even if we allow the baselines to use more than twice as much time, they would only solve a fraction of the problems our method can solve. These results will be added to the paper.
> > >
> > > Regarding batching, all search algorithms we evaluated were implemented with batched best-first search. That is, instead of popping one node OPEN, we pop $k=32$ so that their children can be evaluated in a batch. This is the standard implementation for search algorithms learning a policy and/or heuristic function and all algorithms already benefit from batching.

---

### Official Review · Reviewer_ocyY · 2025-03-11

**Overall Recommendation:** 3

**Summary:**

The paper proposes a new approach for utilizing policy tree search by the inclusion of learned sub-goals. The subgoals are learned online as the tree search expands during a Bootstrap process while attempting to solve problems. One key innovation is the utilization of failed solution trees as data as well (similar to many modern RL algorithms like HER) but by expertly partioning the graph by exploiting the Louvain algorithm. The policy search is hierarchical with the subgoal policy guided by a weighted sample of the high-level subgoal and the low-level subgoal policy. This forms the evaluation function for policy tree search. The subgoal generator uses the VQVAE ( (Vector Quantized Variational Autoencoder). One key distinction is that the search still operates using the problems operator set while the low-level policy guides the search to read the subgoals. Thus, sub-goal representations that do not map to any high-level state are less susceptible to cause problems.

The authors then perform an empirical analysis on 4 domains using the new evaluation function and perform an ablation that showcases the strengths of exploiting failed attempts.

**Claims And Evidence:**

The claims are supported by convincing evidence. Their empirical evaluation showcases that their approach can reduce the computational effort in problems on several domains compared to SOTA baselines.

**Essential References Not Discussed:**

Not applicable

**Experimental Designs Or Analyses:**

The choice of domains is good, baselines are suitable and evaluation metrics are reasonable for the most part although some additional evaluation metrics could be added.

**Methods And Evaluation Criteria:**

Using subgoals to guide search in sparse, binary reward type problems has been a well-known and well-researched topic. The application to policy search in an online fashion and in the hierarchical setting is reasonable.

**Other Comments Or Suggestions:**

N/A

**Other Strengths And Weaknesses:**

1. I think one weakness is the lack of other evaluation metrics like time in seconds on the x-axis. It is quite important to add data with such results especially due to the added computational effort from learning the subgoals is involved.

This also solves the problem of "incomparable" comparisons with HIPS-e since if the node expansions are not directly comparable certainly time is.

2. Another issue is that the analysis of failures is a bit lacking. I would have rather preferred some more detailed analysis on why sokoban seems to perform worse using this approach. Subgoals needing to be undone is pretty common among many problems (eg. Sussmans anomaly) so if your approach cannot be expected to work here then I would question its overall utility. Please expand on why your approach is outperformed by the vanilla methods on Sokoban. It might also make sense to run a quick experiment on Blocksworld (the domain in sussmans anomaly) and analyse whether similar observations can be made.

**Questions For Authors:**

Overall I have a very positive view of this paper. My negative score is due to the lack of clarity regarding W2. I hope the authors can appropriately resolve my questions in weaknesses before I can revise my score.

**Relation To Broader Scientific Literature:**

This work advances the state-of-art in policy guided search algorithms by introducing a method of learning and utilizing learned subgoals online while also exploiting failed attempts.

**Theoretical Claims:**

The theoretical claims that their method enjoys the same guarantees that PHS* provides (Sec 3.5) seems okay to me.

---

> ### Author Rebuttal · Authors · 2025-04-01
>
> Thank you for your helpful comments and feedback.
>
>
> **Weakness 1**
>
>
> We understand the reviewer's concern with running time as some of the gains we have in terms of expansions disappear when solving easier problems. To address this, we ran experiments on a much more difficult version of BoulderDash to highlight the difference in inference speed is not an issue when the problems are too difficult for the baselines. We also increased the network size used in PHS*($\pi$) (denoted LARGE) to match the number of parameters used in PHS*($\pi^{SG}$) to show that simply using more parameters does not help.
>
>
> Our results are summarized in the tables below. Our proposed approach, PHS*($\pi^{SG}$), learned how to solve the difficult set of BoulderDash within 6.25 hours. WA*,PHS*($\pi$), and PHS*($\pi$) LARGE did not learn to solve the problems even when granted 4x more expansions and 2x more time. Increasing the size of the neural network of PHS*($\pi$) does not help either.
>
>
> |Algorithm | Total Loss (Expansions) | Total Time (Hours) | Training Instances Solved | Nodes Generated per Second |
> |---|---:|---:|---:|--:|
> | WA* | 606,000,000 | 15.58 | 16 |1,872.31 |
> | LevinTS($\pi$) | 636,718,638 | 16.69 | 16 | 1,828.51 |
> | PHS*($\pi$) | 599,849,865 | 15.61 | 16 | 1,765.00 |
> | PHS*($\pi$) LARGE | 344,512,985 | 33.04 | 1 | 496.45 |
> | PHS*($\pi^{SG}$) [Ours] | 85,470,564 | 6.25 | 10,000 | 791.69 |
>
>
> We then tested the resulting models on a separate test set of 100 problems to match the procedure done for Table 1. A maximum budget of 512,000 was given for each problem instance. PHS*($\pi^{SG}$) can solve all 100 problems with a bit more than 1000 expansions. The baselines can solve a small fraction of the 100 problems even when granted 100x more expansions.
>
> | Algorithm | Solved | Expansions |
> | --- | --: | --: |
> | WA* | 14 | 272,872.28 |
> | LevinTS($\pi$) | 14 | 280,407.36 |
> | PHS*($\pi$) | 14 | 288,045.64 |
> | PHS*($\pi$) LARGE | 4 | 110,805.75 |
> | PHS*($\pi^{SG}$) [Ours] | 100 | 1,291.37 |
>
> While we can provide the runtime cost of HIPS-e, it would still not be meaningful. We used the open source HIPS-e implementation from the authors, which is implemented in Python. Our experiments and all other baselines are implemented in C++. If we were to compare running time, we would show our approach being substantially faster than HIPS-e, but due to the mismatch in programming languages, this comparison is not meaningful. Note that our comparison to HIPS-e is still meaningful in the sense that while our approach can learn to solve instances of BoulderDash, HIPS-e fails to do so.
>
> In summary, while our proposed approach might not be faster on easier problems, due to the cost of querying more expensive neural models, it can allow us to solve problems we would not be able to solve with the baseline systems considered in our experiments.
>
>
> **Weakness 2**
>
> This is a great point and we are happy we have the chance to address it in our rebuttal. To avoid confusion with the argument that will follow, let us start by saying that our Sokoban results are by no means weak. The method was able to learn how to solve all training instances and performed best in terms of expansions on the test instances. We just did not see the clear advantage in our favor that we see in the other domains. We conjecture that the reason our method performs worse in Sokoban in comparison to the other three problems is due to the training data the clustering algorithm generates. While clustering finds important structures in the other three problems, it did not find helpful structures in Sokoban. For example, in BoulderDash, once the agent unlocks a door, it opens a region of the state space that was not available before. This is the type of structure the clustering algorithm might be capturing. With that, the structure of the underlying state space could be offering subgoals that are helpful in solving these problems. Sokoban might not have such a helpful underlying structure. Note that it is not trivial to verify our conjecture because we do not attempt to reconstruct the subgoals; we use them to condition the policies.
>
> We see the Sokoban results as a demonstration that our approach is robust. Even if the clustering algorithm does not find helpful structures in the state space, we are still able to learn how to solve the problems.
>
> To illustrate that our approach can undo reached subgoals when needed, we collected statistics of when we need to remove boxes from the goal in Sokoban. For the solution paths our method finds, we tracked the statistics of how often a box was pushed onto a goal, then undone (for example, to shuffle around for another box to get through). The table below shows the results. The table shows that in 66% (37+17+9+4) of the problems solved our system had to move at least one box from a goal location.
>
> | Number of times a solved box was undone | Percentage of paths found |
> | --- | ---: |
> | 0 | 33% |
> | 1 | 37% |
> | 2 | 17% |
> | 3 | 9% |
> | 4 | 4%|

---

> > ### Comment · Reviewer_ocyY · 2025-04-06
> >
> > Thanks for your response and additional results. This has resolved my concerns and I have increased my score by 1 point.

---

> > > ### Author Response · Authors · 2025-04-09
> > >
> > > We would like to again thank all the reviewers for their feedback. Also, we would like to thank reviewers **yx2F** and **ocyY** for updating their reviews as a result of our discussions.
> > >
> > > We will be available to answer questions in the next few hours before OpenReview closes, in case reviewers **WUJG** and **nFhe** have any follow-up questions. If we miss the discussion period, we kindly ask them to update their reviews with eventual questions, as their questions will help us improve our work.
> > >
> > > In addition to the results on the more difficult BoulderDash problems, we also completed a run of PHS* with a flat policy $\pi$ and with our subgoal-based policy $\pi^\text{SG}$ on more difficult CraftWorld problems, where the agent needs to craft more complex items. The table below summarizes the results.
> > >
> > > **CraftWorld**
> > > |Algorithm | Total Loss (Expansions) | Total Time (Hours) | Training Instances Solved | Nodes Generated per Second |
> > > |---|---:|---:|---:|--:|
> > > | PHS*($\pi$) | 373,496,109 | 18.35 | 80 | 855.58 |
> > > | PHS*($\pi^\text{SG}$) [Ours] | 123,729,550 | 16.12 | 10,000 | 451.62 |
> > >
> > > Similarly to what we observed in the more difficult problems of BoulderDash, the baseline fails to learn a policy after more than 18 hours of computation, while our system is able to learn an effective policy in 16 hours of computation. We will complete these experiments with all the baselines and include them in the paper.
> > >
> > > We hope that the results on the more difficult BoulderDash and CraftWorld instances will clear the concerns the reviewers had related to the running time of our method and the difficulty of the problems we used in our experiments.

---

### Official Review · Reviewer_nFhe · 2025-03-12

**Overall Recommendation:** 3

**Summary:**

The authors propose a new way to generate policies for deterministic tree search algorithms. Their method to do so is by learning to generate sub-goals using a VQVAE where the subgoals are recognized , learning to reach these subgoals with a low level policy and a high level policy over subgoals.

**Claims And Evidence:**

The experimental results are over 4 domains and looks promising. But the domains looks very similar, and not very large. I wonder how challenging they are, and if the proposed approach is scalable to harder domains and problems.

**Essential References Not Discussed:**

None that I know of.

**Experimental Designs Or Analyses:**

The experimental results in Figure 2 looks sound as far as I can tell.

**Methods And Evaluation Criteria:**

Yes.

**Other Comments Or Suggestions:**

Perhaps you can consider optimization problems as another test-bed (for example TSP and extensions). There are numerous benchmarks there and it should be easy to see if the proposed approach is "competitive".

**Other Strengths And Weaknesses:**

The motivation is good and I think deterministic tree search problems are very common in real world.

The improvements over previous works are substantial and novel.

The paper is a bit complicated and hard to follow with many parts, some text simplication or simpler diagrams of specific components like the Louvain algorithm or the search part would have helped with understanding).

Even though the overall algorithm seems pretty complex, the experiments are done on problems that look pretty small and easy so either an explanation on how hard these problems are, or tackling problems that look harder, would have been a great help to assess the quality of the work.

**Questions For Authors:**

1. How complex are the domains you've experimented on?

2. How well will the method work on much bigger problems?

3. Does the learned sub-tasks in the experiments reach some semantic meaning?

**Relation To Broader Scientific Literature:**

The paper seems to consider related scientific literature appropriately.

**Theoretical Claims:**

No theoretical claims in the paper.

---

> ### Author Rebuttal · Authors · 2025-04-01
>
> Thanks for your helpful comments and feedback.
>
> **Question 1.**
>
> The environment domains chosen are common amongst related works. In terms of complexity, Sokoban is PSPACE-complete [1], CraftWorld and Box-World are NP-hard [2]. BoulderDash requires collecting multiple diamonds before reaching the exit, which is equivalent to the NP-Hard Hamiltonian path problem between the diamonds and the exit of the puzzle. The domains have important differences that make them a good benchmark for search algorithms. Both Sokoban and Box-World have deadlocks (pushing a box into a corner for Sokoban and using a key on the wrong box for Box-World). BoulderDash has a unique property with its dirt elements, where the agent when moving over a dirt tile removes it. This can greatly increase the size of the state-space, and the agent needs to learn that dirt elements can be ignored.
>
> We ran additional experiments on a much larger version of BoulderDash. As before, we use the bootstrap process to train through search WA*, PHS*($\pi$), and our method PHS*($\pi^{SG}$) over 10,000 problems where an initial budget of 4000 is used. The neural networks used for the policy/heuristics in WA* and PHS*($\pi$) have ~2.5M parameters, whereas the total number of parameters for all policies/subgoal generators in PHS*($\pi^{SG}$) is ~10M. We also tried increasing the network size used in PHS*($\pi$) (denoted LARGE) to match the number of parameters used in PHS*($\pi^{SG}$) to show that simply using more parameters does not help.
>
> Our results are summarized in the tables below. Our proposed approach, PHS*($\pi^{SG}$), learned how to solve the difficult set of BoulderDash within 6.25 hours. WA*,PHS*($\pi$), and PHS*($\pi$) LARGE did not learn to solve the problems even when granted 4x more expansions and 2x more time. Increasing the size of the neural network of PHS*($\pi$) does not help either.
>
> |Algorithm | Total Loss (Expansions) | Total Time (Hours) | Training Instances Solved | Nodes Generated per Second |
> |---|---:|---:|---:|--:|
> | WA* | 606,000,000 | 15.58 | 16 |1,872.31 |
> | LevinTS($\pi$) | 636,718,638 | 16.69 | 16 | 1,828.51 |
> | PHS*($\pi$) | 599,849,865 | 15.61 | 16 | 1,765.00 |
> | PHS*($\pi$) LARGE | 344,512,985 | 33.04 | 1 | 496.45 |
> | PHS*($\pi^{SG}$) [Ours] | 85,470,564 | 6.25 | 10,000 | 791.69 |
>
> We then tested the resulting models on a separate test set of 100 problems to match the procedure done for Table 1. A maximum budget of 512,000 was given for each problem instance. PHS*($\pi^{SG}$) can solve all 100 problems with a bit more than 1000 expansions. The baselines can solve a small fraction of the 100 problems even when granted 100x more expansions.
>
> | Algorithm | Solved | Expansions |
> | --- | --: | --: |
> | WA* | 14 | 272,872.28 |
> | LevinTS($\pi$) | 14 | 280,407.36 |
> | PHS*($\pi$) | 14 | 288,045.64 |
> | PHS*($\pi$) LARGE | 4 | 110,805.75 |
> | PHS*($\pi^{SG}$) [Ours] | 100 | 1,291.37 |
>
> **Question 3.**
>
> We are happy to give more details in the paper, but as part of our implementation details we do not use the fully grounded reconstructed observation from the VQVAE decoder. This results in it being difficult to gaininsights to potential semantic meaning of the subgoals produced. HIPS [3] which uses the fully reconstructed observations as their subgoal targets do provide visualizations of the subgoals generated. We do think that this would be an interesting research direction, to see if any semantic meaning can be learned and/or used in downstream tasks.
>
> [1] Culberson, J. Sokoban is PSPACE-complete. IEICE Technical Report, 1997
>
> [2] Viglietta, Giovanni. "Gaming is a hard job, but someone has to do it!." Theory of Computing Systems 54 (2014): 595-621.
>
> [3] Kujanpää, Kalle, Joni Pajarinen, and Alexander Ilin. "Hierarchical imitation learning with vector quantized models." International Conference on Machine Learning. PMLR, 2023.

---

### Official Review · Reviewer_WUJG · 2025-03-14

**Overall Recommendation:** 3

**Summary:**

- The paper proposes an algorithmic framework for best-first search intended to solve deterministic search problems. The primary contributions of the paper are algorithmic and empirical. The main algorithmic idea is to learn control knowledge to speedup search. Subgoals here refer to state abstractions represented by a VQVAE, with the size of the subgoal "space" parameterized by the size of the VQVAE codebook. The resulting control knowledge takes the form of a pair of policies, one generating subgoals and one conditioned on them. These can now be used to guide the search towards states more likely to lead to solutions (based on previously solved problem instances).

- The training algorithm consists of alternating between generating search control knowledge with the current policy and updating the policies and heuristic function in a loop. Key implementation choices include using the Louvain clustering algorithm to generate training data for the subgoal generator (VQVAE) as well as scaling training data for the VQVAE subgoal generator by augmenting the supervised solution sub-trajectories with additional data generated via a clustering-based approach.

- Results on four deterministic search domains show that the proposed algorithm is more sample efficient than the baselines (which do not use the learned control knowledge) without any deterioration in solution quality.

---

## update after rebuttal

  - I thank the authors for their detailed responses to the reviewers. These have addressed my main concerns about the paper. I'm now more positively inclined towards the paper and have revised my score upwards.

**Claims And Evidence:**

- The claims are reasonably well supported.

**Essential References Not Discussed:**

- None that I can spot. The paper's related work section needs some improvement though to make it more complete.

**Experimental Designs Or Analyses:**

- Yes. The experimental setup described in Section 4 and the appendices seems fine to me.

**Methods And Evaluation Criteria:**

- Table 1 shows nearly all baselines and the proposed methods solve all test problem instances. This suggests that the problem domains might be a bit too "easy". It is currently unclear how the method might perform on harder problems.

- The experimental section is somewhat thin. Since search quality is not shown to be improved, it would have been good to see a conclusive empirical demonstration of search speedup. This does not happen in the paper. While number of node expansions is a reasonable metric, I'd suggest a more detailed investigation into runtimes and computational overhead is warranted.

**Other Comments Or Suggestions:**

- None at this time.

**Other Strengths And Weaknesses:**

Strengths

  - The paper tackles an interesting and important problem. The ability to learn search control knowledge for improving search quality and speed, without the need for manual domain engineering, has large utility.
  - The overall approach is intuitively clear.
  - The paper seems to be an interesting and novel combination of prior ideas (Louvain clustering, VQVAE as subgoal generator, best-first search, differentiable policies).
  - The experimental results show that the proposed algorithm needs fewer node expansions to find solutions of similar quality.


Weaknesses

  - The paper mentions its approach in detail but does not offer much motivation or justification for its algorithmic choices. Examples below.
    - What's a formal definition of a subgoal as used in the paper?
    - Why restrict to deterministic problems? What doesn't work if there's randomness or noise in the transition function?
    - What's the intuition for choosing the particular form of $\pi^{SG}$ described in Equation 4? What other forms were considered? Why was this one preferred over those?

  - Considering the large body of work on learning search control knowledge to improve search quality and speed [1,2,3,4], the experimental section needs improvement. It is unclear to me if the proposed method is better than existing techniques. Details follow.
    - All considered domains are solved at test time by all methods including baselines without any improvement in solution quality. This suggests all the domains are "easy" for the baselines. How might the method do in harder domains? The choice of domains needs to be improved in order to answer these questions.
    - The proposed method is an algorithmic framework, with many hyperparameters and choices. Appendix J demonstrates the potential impact of a single poorly chosen hyper-parameter. In my opinion, the paper needs a much more careful empirical investigation with key insights discussed in the main paper.
    - Reporting results in terms of node expansions is not wrong but does mask the overhead of using the control knowledge during search. Does the proposed algorithm actually run faster than the simpler baselines (LevinTS, PHS*)? Please discuss and / or report time-based results. If solution quality was better, this would be less of a concern.
    - [1] Combining Online and Offline Knowledge in UCT (Silver 2007)
    - [2] HC-Search: Learning Heuristics and Cost Functions for Structured Prediction (Doppa 2013)
    - [3] Guided search for task and motion plans using learned heuristics (Chitnis 2016)
    - [4] Learning Simulation Control in General Game-Playing Agents (Finnsson 2010)

  - The related work section could be significantly improved. For example, the kSubS paper contains a more complete discussion of related work. Specifically, the discussion on goal-conditioned RL could be improved with a discussion of hierarchical RL methods (e.g., MAXQ [1], ALISP, options, etc.) and state abstraction [1,2] in particular, given the Louvain algorithm and the k-step training mechanism.
      - [1] Dietterich 1999: https://proceedings.neurips.cc/paper/1999/hash/e5a4d6bf330f23a8707bb0d6001dfbe8-Abstract.html
      - [2] Andre 2002 ; https://cdn.aaai.org/AAAI/2002/AAAI02-019.pdf

  - Overall, due to the lack of improvement in solution quality and limited empirical evaluation, I'm not sure if the paper represents an actual algorithmic improvement over existing methods. While an interesting combination of existing ideas, the overall novelty seems low. As a result, I'm inclined to reject. That said, I do like the overall approach and don't see any major technical issues with the paper, so I'm happy to revise my score upwards based on author feedback.

**Questions For Authors:**

- (Q1) Based on Table 1, it seems nearly all baselines and proposed methods solve all test problem instances. Might this suggest that the selected domains are too "easy"? What might be more challenging domains and how might the results change on these domains?

- (Q2) Is LevinTS($\pi^{SG}$) actually faster than LevinTS in wall-clock time? Please provide details about actual runtime at test time for all the methods considered. I'm looking to better understand how much search speedup is achieved by using the VQVAE and hierarchical policy.

**Relation To Broader Scientific Literature:**

- The main algorithmic contribution is a good combination of ideas from prior work (Louvain clustering, VQVAE-based subgoal generation, k-step data generation as shown in kSubS and AdaSubS).

**Theoretical Claims:**

- The paper does not focus on theoretical aspects. The properties of the search seem fine to me.

---

> ### Author Rebuttal · Authors · 2025-04-01
>
> Thank you for your helpful comments and feedback.
>
> **Question 1**
>
> Thank you for your suggestion to run on more difficult instances. We ran additional experiments on a much larger version of BoulderDash, following the same procedure from **Section 4.3**. We also tried increasing the network size used in PHS*($\pi$) (denoted LARGE) to match the number of parameters used in PHS*($\pi^{SG}$) to show that simply using more parameters does not help.
>
> Our results are summarized in the tables below. Our proposed approach, PHS*($\pi^{SG}$), learned how to solve the difficult set of BoulderDash within 6.25 hours. The baselines did not learn to solve the problems even when granted 4x more expansions and 2x more time.
>
> |Algorithm | Total Loss (Expansions) | Total Time (Hours) | Training Instances Solved | Nodes Generated per Second |
> |---|---:|---:|---:|--:|
> | WA* | 606,000,000 | 15.58 | 16 |1,872.31 |
> | LevinTS($\pi$) | 636,718,638 | 16.69 | 16 | 1,828.51 |
> | PHS*($\pi$) | 599,849,865 | 15.61 | 16 | 1,765.00 |
> | PHS*($\pi$) LARGE | 344,512,985 | 33.04 | 1 | 496.45 |
> | PHS*($\pi^{SG}$) [Ours] | 85,470,564 | 6.25 | 10,000 | 791.69 |
>
> We then tested the resulting models on a separate test set of 100 problems to match the procedure done for Table 1.
> | Algorithm | Solved | Expansions |
> | --- | --: | --: |
> | WA* | 14 | 272,872.28 |
> | LevinTS($\pi$) | 14 | 280,407.36 |
> | PHS*($\pi$) | 14 | 288,045.64 |
> | PHS*($\pi$) LARGE | 4 | 110,805.75 |
> | PHS*($\pi^{SG}$) [Ours] | 100 | 1,291.37 |
>
> **Question 2**
>
> The results on the more difficult instances of BoulderDash we presented above highlight that PHS*($\pi^{SG}$) can be substantially faster than the baselines, to the point where PHS*($\pi^{SG}$) learns how to solve the problems while the baselines do not.  Next, we present the runtime statistics for the methods listed in Table 1, on the easier problem instances, for a representative example; the tables for the other domains are similar and were omitted for space.
>
> What the table below shows is that, from a practical perspective, all algorithms tested, once they learn a policy and/or heuristic, are able to solve the test problems within a reasonable amount of time. The key difference between the method we propose and the baselines is the time that it might take them to learn the policy and heuristic. As demonstrated in the more difficult BoulderDash instances, the difference between our approach and baselines can be the same as solving all instances versus solving none of the instances.
>
> | Algorithm | Solved | Expansions | Length | Total Time (Seconds) |
> | --- | ---: | ---: | ---: | ---: |
> |  | | **BoulderDash** | | |
> | WA* (1.5) | 100 | 1,193.60 | 51.44 | 31.93 |
> | LevinTS($\pi$) | 100 | 61.33 | 52.90 | 4.82 |
> | PHS*($\pi$) | 100 | 53.65 | 52.74 | 5.60 |
> | LevinTS($\pi^{SG}$) | 100 | 65.48 | 53.30 | 16.18 |
> | PHS*($\pi^{SG}$) | 100 | 53.34 | 52.68 | 10.89 |
>
> The following addresses your points from the Weaknesses section.
> - **Formal definition of a subgoal**: Subgoals are states from the underlying state space that the search attempts to achieve.  We will clarify this in the paper
> - **Restricting to deterministic problems**: Deterministic problems are an important and active area of research. There is also a body of work which looks at how transformations can be applied to non-classical planning problems (e.g., non-determinism/incomplete information) from the classical planning setting we consider in our work [5]
> - **Is the proposed method is better than existing technique**: WA*, LevinTS, and PHS* are the current state of the art methods for the problems we consider. [1] and [4] consider two-player games and Orseau & Lelis showed that PUCT performs quite poorly on this type of problem. [3] solves MDPs, whereas we consider the classical planning setting. [2] requires a dataset of structured input/outputs, whereas we focus on needle-in-a-haystack problems. The suggested references, [1-4], are only related to our work in a broad sense.
> - **Intuition for Equation (4)**: We provide the intuition of Equation 4 with an example. Consider the case where one subgoal could be "go to the door on left", while another could be "go to the door on the right". Mixing these two subgoals would result in a uniform (and ineffective) policy. The high-level policy decides which subgoal to attain next by providing a weight to the probability distribution given by the low-level policies. This way, the high-level policy decides whether the agent will go right or left.
> - **Baselines do not use the learned control knowledge**: All baselines learn a policy and/or heuristic. In particular, LevinTS/PHS* use the same Bootstrap pipeline that our method uses.
>
> [5] Geffner, Hector. "Non-classical planning with a classical planner: The power of transformations." European Workshop on Logics in Artificial Intelligence. Cham: Springer International Publishing, 2014.

---

### Decision · Program_Chairs · 2025-05-01

**Decision:**

Accept (poster)

**Comment:**

This paper proposes a subgoal-guided policy tree search with learned subgoals. Unlike previous approaches where subgoals are learned from trajectories, the proposed method learns subgoals online using a VQVAE based on the search tree. Experiments show that the proposed method improves sample efficiency in online settings.

The paper provides a new perspective on incorporating subgoal planning, which is an important topic. The experimental results support the claims and are promising. All reviewers agree it is a good paper. Overall, I recommend acceptance.